# DISCRETE REPRESENTATIONS STRENGTHEN VISION TRANSFORMER ROBUSTNESS

**Chengzhi Mao**[2]*, **Lu Jiang**[1], **Mostafa Dehghani**[1], **Carl Vondrick**[2], **Rahul Sukthankar**[1], **Irfan Essa**[1,3]
[1] Google Research
[2] Computer Science, Columbia University
[3] School of Interactive Computing, Georgia Insitute of Technology
`{mcz, vondrick}@cs.columbia.edu`
`{lujiang, dehghani, sukthankar, irfanessa}@google.com`

## ABSTRACT

Vision Transformer (ViT) is emerging as the state-of-the-art architecture for image recognition. While recent studies suggest that ViTs are more robust than their convolutional counterparts, our experiments find that ViTs trained on ImageNet are overly reliant on local textures and fail to make adequate use of shape information. ViTs thus have difficulties generalizing to out-of-distribution, real-world data. To address this deficiency, we present a simple and effective architecture modification to ViT's input layer by adding discrete tokens produced by a vector-quantized encoder. Different from the standard continuous pixel tokens, discrete tokens are invariant under small perturbations and contain less information individually, which promote ViTs to learn global information that is invariant. Experimental results demonstrate that adding discrete representation on four architecture variants strengthens ViT robustness by up to 12% across seven ImageNet robustness benchmarks while maintaining the performance on ImageNet.

## 1 INTRODUCTION

Despite their high performance on in-distribution test sets, deep neural networks fail to generalize under real-world distribution shifts (Barbu et al., 2019). This gap between training and inference poses many challenges for deploying deep learning models in real-world applications where closed-world assumptions are violated. This lack of robustness can be ascribed to learned representations that are overly sensitive to minor variations in local texture and insufficiently adept at representing more robust scene and object characteristics, such as the shape.

Vision Transformer (ViT) (Dosovitskiy et al., 2020) has started to rival Convolutional Neural Networks (CNNs) in many computer vision tasks. Recent works found that ViTs are more robust than CNNs (Paul & Chen, 2021; Mao et al., 2021b; Bhojanapalli et al., 2021) and generalize favorably on a variety of visual robustness benchmarks (Hendrycks et al., 2021b; Hendrycks & Dietterich, 2019). These work suggested that ViTs' robustness comes from the self-attention architecture that captures a globally-contextualized inductive bias than CNNs.

However, though self-attention can model the shape information, we found that ImageNet-trained ViT is still biased to textures than shape (Geirhos et al., 2019). We hypothesize that this deficiency in robustness comes from the high-dimensional, individually informative, linear tokenization which biases ViT to minimize empirical risk via local signals without learning much shape information.

In this paper, we propose a simple yet novel input layer for vision transformers, where image patches are represented by *discrete tokens*. To be specific, we discretize an image and represent an image patch as a *discrete token* or "visual word" in a codebook. Our key insight is that discrete tokens capture important features in a low-dimensional space (Oord et al., 2017) preserving shape and structure of the object (see Figure 2). Our approach capitalizes on this discrete representation to promote the robustness of ViT. Using discrete tokens drives ViT towards better modeling of spatial interactions between tokens, given that individual tokens no longer carry enough information to

---

*Work done while interning at Google Research.

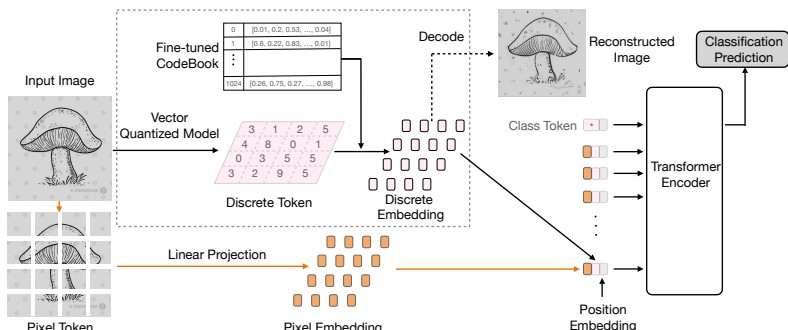

Figure 1: Overview of the proposed ViT using discrete representations. In addition to the pixel embeddings (orange), we introduce discrete tokens and embeddings (pink) as the input to the standard Transformer Encoder of the ViT model (Dosovitskiy et al., 2020).

depend on. We also concatenate a low dimensional pixel token to the discrete token to compensate for the potentially missed local details encoded by discrete tokens, especially for small objects.

Our approach only changes the image patch tokenizer to improve generalization and robustness, which is orthogonal to all existing approaches for robustness, and can be integrated into architectures that extend vision transformer. We call the ViT model using our *Discrete representation* or *Dr.* ViT.

Our experiments and visualizations show that incorporating discrete tokens in ViT significantly improves generalization accuracy for all seven out-of-distribution ImageNet benchmarks: Stylized-ImageNet by up to 12%, ImageNet-Sketch by up to 10%, and ImageNet-C by up to 10%. Our method establishes the new state-of-the-art on four benchmarks without any dataset-specific data augmentation. The success of discrete representation has been limiting to image generation (Ramesh et al., 2021; Esser et al., 2021). Our work is the first to connect discrete representations to robustness and demonstrate consistent robustness improvements. We hope our work helps pave the road to a joint vision transformer for both image classification and generation.

## 2 RELATED WORK

**Vision Transformer (ViT)** (Dosovitskiy et al., 2020), inspired by the Transformer (Vaswani et al., 2017) in NLP, is the first CNN-free architecture that achieves state-of-the-art image classification accuracy. Since its inception, numerous works have proposed improvements to the ViT architecture (Wang et al., 2021; Chu et al., 2021; Liu et al., 2021; d'Ascoli et al., 2021), objective (Chen et al., 2021a), training strategy (Touvron et al., 2021), etc.. Given the difficulty to study all existing ViT models, this paper focuses on the classical ViT model (Dosovitskiy et al., 2020) and its recent published versions. Specifically, Steiner et al. (2021) proposed ViT-AugReg that applies stronger data augmentation and regularization to the ViT model. Tolstikhin et al. (2021) introduced MLP-Mixer to replace self-attention in ViT with multi-layer perceptions (MLP).

We select the above ViT model family (Dosovitskiy et al., 2020; Steiner et al., 2021; Tolstikhin et al., 2021) in our robustness study for three reasons. First, they represent both the very first and one-of-the-best vision transformers in the literature. Second, these models demonstrated competitive performance when pre-trained on sufficiently large datasets such as ImageNet-21K and JFT-300M. Finally, unlike other Transformer models, they provide architectures consisting of solely Transformer layers as well as a hybrid of CNN and Transformer layers. These properties improve our understanding of robustness for different types of network layers and datasets.

**Robustness.** Recent works established multiple content robustness datasets to evaluate the out-of-distribution generalization of deep models (Barbu et al., 2019; Hendrycks et al., 2021b;a; Wang et al., 2019; Geirhos et al., 2019; Hendrycks & Dietterich, 2019; Recht et al., 2019). In this paper, we consider 7 ImageNet robustness benchmarks of real-world test images (or proxies) where deep models trained on ImageNet are shown to suffer from notable performance drop. Existing works on robustness are targeted at closing the gap in a subset of these ImageNet robustness benchmarks and were extensively verified with the CNNs. Among them, carefully-designed data augmentations (Hendrycks et al., 2021a; Cubuk et al., 2018; Steiner et al., 2021; Mao et al., 2021b;a), model regularization (Wang et al., 2019; Huang et al., 2020b; Hendrycks et al., 2019), and multitask learning (Zamir et al., 2020) are effective to address the issue.

More recently, a few studies (Paul & Chen, 2021; Bhojanapalli et al., 2021; Naseer et al., 2021; Shao et al., 2021) suggest that ViTs are more robust than CNNs. Existing works mainly focused

on analyzing the cause of superior generalizability in the ViT model. As our work focuses on discrete token input, we train the models using the same data augmentation as the ViT-AugReg baseline (Steiner et al., 2021). While tailoring data augmentation (Mao et al., 2021b) may further improve our results, we leave it out of the scope of this paper.

**Discrete Representation** was used as a visual representation prior to the deep learning revolution, such as in bag-of-visual-words model (Sivic & Zisserman, 2003; Csurka et al., 2004) and VLAD model (Arandjelovic & Zisserman, 2013). Recently, (Oord et al., 2017; Vahdat et al., 2018) proposed neural discrete representation to encode an image as integer tokens. Recent works used discrete representation mainly for image synthesis (Ramesh et al., 2021; Esser et al., 2021). To the best of our knowledge, our work is the first to demonstrate discrete representations strengthening robustness. The closest work to ours is BEiT (Bao et al., 2021) that pretrains the ViTs to predict the masked tokens. However, the tokens are discarded after pretraining, where the ViT model can still overfit the non-robust nuisances in the pixel tokens at later finetuning stage, undermining its robustness.

## 3 METHOD

### 3.1 PRELIMINARY ON VISION TRANSFORMER

Vision Transformer (Dosovitskiy et al., 2020) is a pure transformer architecture that operates on a sequence of image patches. The 2D image $\mathbf{x} \in \mathbb{R}^{H \times W \times C}$ is flattened into a sequence of image patches, following the raster scan, denoted by $\mathbf{x}_p \in \mathbb{R}^{L \times (P^2 \cdot C)}$, where $L = \frac{H \times W}{P^2}$ is the effective sequence length and $P^2 \times C$ is the dimension of image patch. A learnable classification token $\mathbf{x}_{\text{class}}$ is prepended to the patch sequence, then the position embedding $\mathbf{E}_{pos}$ is added to formulate the final input embedding $\mathbf{h}_0$.

$$\mathbf{h}_0 = [\mathbf{x}_{\text{class}}; \mathbf{x}_p^1 \mathbf{E}; \mathbf{x}_p^2 \mathbf{E}; \cdots ; \mathbf{x}_p^L \mathbf{E}] + \mathbf{E}_{pos}, \qquad \mathbf{E} \in \mathbb{R}^{(P^2 \cdot C) \times D}, \mathbf{E}_{pos} \in \mathbb{R}^{(L+1) \times D} \quad (1)$$

$$\mathbf{h}'_\ell = \text{MSA}(\text{LN}(\mathbf{h}_{\ell-1})) + \mathbf{h}_{\ell-1}, \qquad \ell = 1, \ldots, L_f \quad (2)$$

$$\mathbf{h}_\ell = \text{MLP}(\text{LN}(\mathbf{h}'_\ell)) + \mathbf{h}'_\ell, \qquad \ell = 1, \ldots, L_f \quad (3)$$

$$\mathbf{y} = \text{LN}(\mathbf{h}_L^0), \quad (4)$$

The architecture of ViT follows that of the Transformer (Vaswani et al., 2017), which alternates layers of multi-headed self-attention (MSA) and multi-layer perceptron (MLP) with LayerNorm (LN) and residual connections being applied to every block. We denote the number of blocks as $L_f$.

This paper considers the ViT model family consisting of 4 ViT backbones: the vanilla ViT discussed above, ViT-AugReg (Steiner et al., 2021) which shares the same ViT architecture but applies stronger data augmentation and regularization, MLP-Mixer (Tolstikhin et al., 2021) which replaces self-attention in ViT with MLP, a variant called Hybrid-ViT which replaces the raw image patches in Equation 1 with the CNN features extracted by a ResNet-50 (He et al., 2016).

### 3.2 ARCHITECTURE

Existing ViTs represent an image patch as a sequence of *pixel tokens*, which are linear projections of flattened image pixels. We propose a novel architecture modification to the input layer of the vision transformer, where an image patch $\boldsymbol{x}_p$ is represented by a combination of two embeddings. As illustrated in Fig. 1, in addition to the original pixel-wise linear projection, we discretize an image patch into an *discrete token* in a codebook $\mathbf{V} \in \mathbb{R}^{K \times d_c}$, where $K$ is the codebook size and $d_c$ is the dimension of the embedding. The discretization is achieved by a vector quantized (VQ) encoder $p_\theta$ that produces an integer $z$ for an image patch $x$ as:

$$p_\theta(z = k|x) = \mathbf{1}(k = \underset{j=1:K}{\arg\min} \|z_e(x) - \mathbf{V}_j\|_2), \quad (5)$$

where $z_e(x)$ denotes the output of the encoder network and $\mathbf{1}(\cdot)$ is the indicator function.

The encoder is applied to the patch sequence $\boldsymbol{x}_p \in \mathbb{R}^{L \times (P^2 \cdot C)}$ to obtain an integer sequence $\boldsymbol{z}_d \in \{1, 2, ..., K\}^L$. Afterward, we use the embeddings of both discrete and pixel tokens to construct the input embedding to the ViT model. Specifically, the input embedding in Equation 1 is replaced by:

$$\mathbf{h}_0 = [\mathbf{x}_{\text{class}}; f(\mathbf{V}_{\mathbf{z}_d^1}, \mathbf{x}_p^1 \mathbf{E}); f(\mathbf{V}_{\mathbf{z}_d^2}, \mathbf{x}_p^2 \mathbf{E}); \cdots ; f(\mathbf{V}_{\mathbf{z}_d^L}, \mathbf{x}_p^L \mathbf{E})] + \mathbf{E}_{pos}, \quad (6)$$

where $f$ is the function, embodied as a neural network layer, to combine the two embeddings. We empirically compared four network designs for $f$ and found that the simplest concatenation works

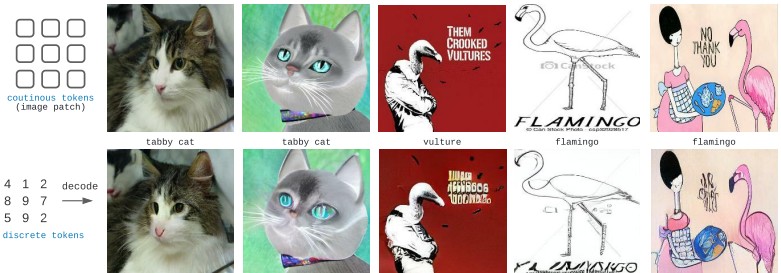

Figure 2: Comparison of pixel tokens (top) and the reconstructed image decoded from the discrete tokens (bottom). Discrete tokens capture important shapes and structures but may lose local texture.

best. Note that our model only modifies the input layer of ViT (Equation 1) and leaves intact the remaining layers depicted in Equation 2-4.

**Comparison of pixel and discrete embeddings:** Pixel and discrete embeddings represent different aspects of the input image. *Discrete embeddings* capture important features in a low-dimension space (Oord et al., 2017) that preserves the global structure of an object but lose local details. Fig. 2 compares the original image (top) and the reconstructed images decoded from the discrete embeddings of our model (bottom). As shown, the decoded images from discrete embeddings reasonably depict the object shape and global context. Due to the quantization, the decoder hallucinates the local textures, e.g., in the cat's eye, or the text in the "vulture" and "flamingo" images. It is worth noting that the VQ encoder/decoder is only trained on ImageNet 2012 but they can generalize to out-of-distribution images. Please see more examples in Appendix A.

On the flip side, *pixel embeddings* capture rich details through the linear projection from raw pixels. However, given the expressive power of transformers, ViTs can spend capacity on local textures or nuance patterns that are often circumferential to robust recognition. Since humans recognize images primarily relying on the shape and semantic structure, this discrepancy to human perception undermines ViT's generalization on out-of-distribution data. Our proposed model leverages the power of both embeddings to promote the interaction between modeling global and local features.

### 3.3 TRAINING PROCEDURE

The training comprises two stages: pretraining and finetuning. First, we pretrain the VQ-VAE encoder and decoder (Oord et al., 2017) on the given training set. We do not use labels in this step. In the finetuning stage, as shown in Fig. 3a, we train the proposed ViT model from scratch and finetune the discrete embeddings. Due to the straight-through gradient estimation in VQ-VAE (Oord et al., 2017), we stop the back-propagation after the discrete embedding.

```
1  # x: input image mini-batch; pixel_embed: pixel
2  # embeddings of x.  vqgan.encoder and codebook are
3  # initialized form the pretraining.
4  import jax.numpy as np
5  discrete_token = jax.lax.stop_gradient(vqgan.encoder(x))
6  discrete_embed = np.dot(discrete_token, codebook)
7  tokens = np.concatenate(
8          [discrete_embed, pixel_embed], dim=2)
9  predictions = TransformerEncoder(tokens)
```

(a) Pseudo JAX code

(b) Pixel/RGB embedding filters

Figure 3: (a) Pseudo code for training the proposed ViT model. (b) Comparing visualized pixel embeddings of the ViT and our model. Top row shows the randomly selected filters and Bottom shows the first 28 principal components. Our filters capture more structural and shape patterns.

Now we discuss the objective that the proposed ViT model optimizes. Let $q_\phi(z|x)$ denote the VQ encoder, parameterized by network weights $\phi$, to represent an image $x$ into a sequence of integer tokens $z$. The decoder models the distribution $p_\theta(x|z)$ over the RGB image generated from discrete tokens. $p_\psi(y|x,z)$ stands for the proposed vision transformer shown in Fig. 1.

We factorize the joint distribution of image $x$, label $y$, and the discrete token $z$ by $p(x,y,z) = p_\phi(x|z)p_\psi(y|x,z)p(z)$. Our overall training procedure is to maximize the evidence lower bound

(ELBO) on the joint likelihood:

$$\log p(x, y) \geq \mathbb{E}_{q_\phi(z|x)}[\log p_\theta(x|z)] - D_{\mathrm{KL}}[q_\phi(z|x) \| p_\psi(y|x, z)p(z)] \tag{7}$$

In the first stage, we maximize the ELBO with respect to $\phi$ and $\theta$, which corresponds to learning the VQ-VAE encoder and decoder. Following (Oord et al., 2017), we assume a uniform prior for both $p_\psi(y|x, z)$ and $p(z)$. Given that $q_\phi(z|x)$ predicts a one-hot output, the regularization term (KL divergence in Equation 7) will be a constant. Note that the DALL-E model (Ramesh et al., 2021) uses a similar assumption to stabilize the training. Our implementation uses VQ-GAN (Esser et al., 2021) which adds a GAN loss and a perceptual loss. We also include results with VQ-VAE (Oord et al., 2017) in Appendix A.8 for reference.

In the finetuning stage, we optimize $\psi$ while holding $\theta$ and $\phi$ fixed, which corresponds to learning a ViT and finetuning the discrete embeddings. This can be seen by rearranging the ELBO:

$$\log p(x, y) \geq \mathbb{E}_{q_\phi(z|x)}[\log p_\psi(y|x, z)] + \mathbb{E}_{q_\phi(z|x)}[\log \frac{p_\theta(x|z)p(z)}{q_\phi(z|x)}], \tag{8}$$

where $p_\psi(y|x, z)$ is the proposed ViT model. The first term denotes the likelihood for classification that is learned by minimizing the multi-class cross-entropy loss. The second term becomes a constant given the fixed $\theta$ and $\phi$. It is important to note that the discrete embeddings are learned end-to-end in both pretraining and finetuning. Details are discussed in Appendix A.2.

## 4 EXPERIMENTS

### 4.1 EXPERIMENTAL SETUP

**Datasets.** In the experiments, we train all the models, including ours, on *ImageNet 2012* or *ImageNet-21K* under the same training settings, where we use identical training data, batch size, and learning rate schedule, etc. Afterward, the trained models are tested on the ImageNet robustness benchmarks to assess their robustness and generalization capability.

In total, we evaluate the models on nine benchmarks. ImageNet and ImageNet-Real are two in-distribution datasets. ***ImageNet*** (Deng et al., 2009) is the standard validation set of ILSVRC2012. *ImageNet-**Real*** (Beyer et al., 2020) corrects the label errors in the ImageNet validation set (Northcutt et al., 2021), which measures model's generalization on different labeling procedures.

Seven out-of-distribution (OOD) datasets are considered. Fig. 4 shows their example images. *ImageNet-**Rendition*** (Hendrycks et al., 2021a) is an OOD dataset that contains renditions, such as art, cartoons, of 200 ImageNet classes. ***Stylized**-ImageNet* (Geirhos et al., 2019) is used to induce the texture vs. the shape bias by stylizing the ImageNet validation set with 79,434 paintings. The "shape ImageNet labels" are used as the label. *ImageNet-**Sketch*** (Wang et al., 2019) is a "black and white" dataset constructed by querying the sketches of the 1,000 ImageNet categories. ***Object-Net*** (Barbu et al., 2019) is an OOD test set that controls the background, context, and viewpoints of the data. Following the standard practice, we evaluate the performance on the 113 overlapping ImageNet categories. *ImageNet-**V2*** (Recht et al., 2019) is a new and more challenging test set collected for ImageNet. The split "matched-frequency" is used. *ImageNet-**A*** (Hendrycks et al., 2021b) is a natural adversarial dataset that contains real-world, unmodified, natural examples that cause image classifiers to fail significantly. *ImageNet-**C*** (Hendrycks & Dietterich, 2019) evaluates the model's robustness under common corruptions. It contains 5 serveries of 15 synthetic corruptions including 'Defocus,' 'Fog,' and 'JPEG compression'.

**Models.** We verify our method on three ViT backbones: the classical ViT (Dosovitskiy et al., 2020) termed as ViT Vanilla, ViT-AugReg (Steiner et al., 2021), and MLPMixer (Tolstikhin et al., 2021). By default, we refer ViT to ViT-AugReg considering its superior performance. For ViT, we study the Tiny (Ti), the Small (S), and the Base (B) variants, all using the patch size of 16x16. We also compare with a CNN-based ResNet (He et al., 2016) baseline, and the Hybrid-ViT model (Dosovitskiy et al., 2020) that replaces image patch embedding with a ResNet-50 encoder.

**Implementation details** On *ImageNet*, the models are trained with three ViT model variants, i.e. Ti, S, and B, from small to large. The codebook size is $K = 1024$, and the codebook embeds the discrete token into $d_c = 256$ dimensions. On *ImageNet-21K*, the quantizer model is a VQGAN and is trained on ImageNet-21K only with codebook size $K = 8,192$. All models including ours use the same augmentation (RandAug and Mixup) as in the ViT baseline (Steiner et al., 2021). More details can be found in the Appendix A.12.

Table 1: Model performance trained on ImageNet. All columns indicate the top-1 classification accuracy except the last column ImageNet-C which indicates the mean Corruption Error (the lower the better). The bold number indicates higher accuracy than the corresponding baseline. The box highlights the best accuracy.

| Model | ImageNet | Real | Rendition | Stylized | Sketch | ObjectNet | V2 | A | C ↓ |
|---|---|---|---|---|---|---|---|---|---|
| | | | | | | Out of Distribution Robustness Test | | | |
| ResNet-50 | 76.61 | 83.01 | 36.35 | 6.56 | 23.06 | 26.52 | 64.70 | 4.83 | 75.07 |
| ViT-B Vanilla | 72.47 | 78.69 | 24.56 | 5.94 | 14.34 | 13.44 | 59.32 | 5.23 | 88.72 |
| + Ours (discrete only) | **73.73** | **79.63** | **34.61** | **8.94** | **23.66** | **20.84** | **60.30** | **6.45** | **74.82** |
| ViT-Ti | 58.75 | 66.30 | 21.37 | 6.17 | 10.76 | 12.63 | 46.89 | 3.73 | 86.62 |
| +Ours (discrete only) | **61.74** | **69.94** | **32.35** | **13.44** | **19.94** | **15.35** | **50.60** | **3.81** | **83.62** |
| MLPMixer | 68.33 | 74.74 | 30.65 | 7.03 | 21.54 | 13.47 | 53.52 | 5.20 | 81.11 |
| + Ours (discrete only) | 68.00 | 74.36 | **33.85** | **8.98** | **25.36** | **15.44** | **54.12** | 4.88 | **80.75** |
| + Ours | **69.23** | **76.04** | **36.27** | **13.05** | **26.34** | **16.45** | **55.68** | 4.84 | **72.06** |
| Hybrid ViT-S | 75.10 | 81.45 | 34.01 | 7.42 | 26.69 | 24.50 | 62.53 | 6.17 | 69.26 |
| ViT-S | 75.21 | 82.36 | 34.39 | 10.39 | 23.27 | 24.50 | 63.00 | 10.39 | 61.99 |
| + Ours (discrete only) | 72.42 | 80.14 | **42.58** | **18.44** | **31.16** | 23.95 | 60.89 | **7.52** | 66.82 |
| + Ours | **77.03** | **83.59** | **39.02** | **14.22** | **28.78** | **26.49** | **64.49** | **11.85** | **56.89** |
| Hybrid ViT-B | 74.94 | 80.54 | 33.03 | 7.50 | 25.33 | 23.08 | 61.30 | 7.44 | 69.61 |
| ViT-B | 78.73 | 84.85 | 38.15 | 10.39 | 28.60 | 28.71 | 67.34 | 16.92 | 53.51 |
| + Ours (discrete only) | 78.67 | 84.28 | **48.82** | **22.19** | **39.10** | **30.27** | 66.52 | 14.77 | 55.21 |
| + Ours | **79.48** | **84.86** | **44.77** | **19.38** | **34.59** | **30.55** | **68.05** | **17.20** | **46.22** |
| ViT-B (384x384) | 81.63 | 85.06 | 38.23 | 7.58 | 28.07 | 32.36 | 68.57 | 24.01 | 59.01 |
| + Ours | **81.83** | **86.48** | **44.70** | **14.06** | **35.72** | **36.01** | **70.33** | **27.19** | **46.32** |

## 4.2 MAIN RESULTS

Table 1 shows the results of the models trained on ImageNet. All models are trained under the same setting with the same data augmentation from (Steiner et al., 2021) except for the ViT-B Vanilla row, which uses the data augmentation from (Dosovitskiy et al., 2020). Our improvement is entirely attributed to the proposed discrete representations. By adding discrete representations, all ViT variants, including Ti, S, and B, improve robustness across all eight benchmarks. When only using discrete representations as denoted by "(discrete only)", we observe a larger margin of improvement (**10%-12%**) on the datasets depicting object shape: Rendition, Sketch, and Stylized-ImageNet. It is worth noting that it is a challenging for a single model to obtain robustness gains across all the benchmarks as different datasets may capture distinct types of data distributions.

Table 2: Model performance when pretrained on ImageNet21K and finetuned on ImageNet with 384x384 resolution. See the caption of Table 1 for more description.

| Model | ImageNet | Real | Rendition | Stylized | Sketch | ObjectNet | V2 | A | C ↓ |
|---|---|---|---|---|---|---|---|---|---|
| | | | | | | Out of Distribution Robustness Test | | | |
| ViT-B | 84.20 | 88.61 | 51.23 | 13.59 | 37.83 | 44.30 | 74.54 | 46.59 | 49.62 |
| + Ours (discrete only) | 83.40 | 88.44 | **55.26** | **19.69** | **44.72** | 43.92 | 72.68 | 36.64 | **45.86** |
| + Ours | **84.43** | **88.88** | **54.64** | **18.05** | **41.91** | **44.61** | **75.17** | 44.97 | **38.74** |
| + Ours (512x512) | **85.07** | **89.04** | **54.27** | **16.02** | **41.92** | **46.62** | **75.55** | **52.64** | **38.97** |

Our approach scales as more training data is available. Table 2 shows the performance for the models pretrained on ImageNet-21K and finetuned on ImageNet. Training on sufficiently large datasets inherently improves robustness (Bhojanapalli et al., 2021) but also renders further improvement even more challenging. Nevertheless, our model consistently improves the baseline ViT-B model across all robustness benchmarks (by up to 10% on ImageNet-C). The results in Table 1 and Table 2 validate our model as a generic approach that is highly effective for the models pretrained on the sufficiently large ImageNet-21k dataset.

**Comparison to the State-of-the-Art** We compare our model with the state-of-the-art results, including ViT with CutOut (DeVries & Taylor, 2017) and CutMix (Yun et al., 2019), on four datasets in Table 3 and Table 4. It is noteworthy that different from the compared approaches that are tailored to specific datasets, our method is generic and uses the same data augmentation as our ViT baseline (Steiner et al., 2021), i.e. RandAug and Mixup, for all datasets.

On *ImageNet-Rendition*, we compare with the leaderboard numbers from (Hendrycks et al., 2021a). Trained on ImageNet, our approach beats the state-of-the-art DeepAugment+AugMix approach by

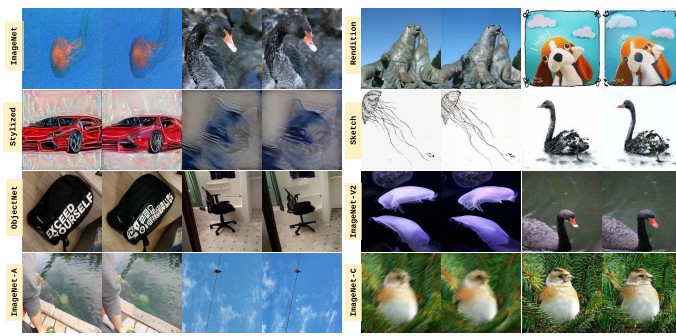

Figure 4: Visualization of the eight evaluation benchmarks. Each image consists of the original test image (Left) and the decoded image from the finetuned discrete embeddings (Right). Note that the encoder and decoder are trained only on ImageNet 2012 data but generalize on out-of-distribution datasets. See more examples in Appendix A.1.1.

Table 3: Comparison to the state-of-the-art classification accuracy on three ImageNet robustness datasets.

| Model | Rendition |
|---|---|
| ResNet 50 | 36.1 |
| ResNet 50 *21K | 37.2 |
| DeepAugment | 42.2 |
| DeepAugment + AugMix | 46.8 |
| RandAug + Mixup | 29.6 |
| ViT-B | 38.2 |
| ViT-B + CutOut | 38.1 |
| ViT-B + CutMix | 38.4 |
| Our ViT-B (discrete only) | **48.8** |
| Our ViT-B *21K | **55.3** |

(a) ImageNet-Rendition

| Model | Sketch |
|---|---|
| Huang et al. (2020a) | 16.1 |
| RandAug + Mixup | 17.7 |
| Xu et al. (2020) | 18.1 |
| Mishra et al. (2020) | 24.5 |
| Hermann et al. (2020) | 30.9 |
| ViT-B | 23.3 |
| ViT-B + CutOut | 26.9 |
| ViT-B + CutMix | 27.5 |
| Our ViT-B (discrete only) | **39.1** |
| Our ViT-B (discrete only) *21K | **44.7** |

(b) ImageNet-Sketch

| Model | Stylized Top5 |
|---|---|
| BagNet-9 | 1.4 |
| BagNet-17 | 2.5 |
| BagNet-33 | 4.2 |
| ResNet-50 | 16.4 |
| ViT-B | 22.2 |
| ViT-B + CutOut | 24.7 |
| ViT-B + CutMix | 22.7 |
| ViT-B *21K | 31.3 |
| Our ViT-B (discrete only) | **40.3** |

(c) Stylized-ImageNet

2%. Notice that the augmentation we used—RandAug+MixUp—is 17% worse than the DeepAugment+AugMix on ResNets. Our performance can be further improved by another 6% when pretrained on ImageNet-21K. On *ImageNet-Sketch*, we surpass the state-of-the-art (Hermann et al., 2020) by 8%. On *Stylized-ImageNet*, our approach improves 18% top-5 accuracy by switching to discrete representation. On *ImageNet-C*, our approach slashes the prior mCE by up to 10%. Note that most baselines are trained with CNN architectures that have a smaller capacity than ViT-B. The comparison is fair for the bottom entries that are all built on the same ViT-B backbone.

### 4.3 IN-DEPTH ANALYSIS

In this subsection, we demonstrate, both quantitatively and qualitatively, that discrete representations facilitate ViT to better capture object shape and global contexts.

**Quantitative results.** Result in Table 3c studies the OOD generalization setting "IN-SIN" used in (Geirhos et al., 2019), where the model is trained only on the ImageNet (IN) and tested on the Stylized ImageNet (SIN) images with conflicting shape and texture information, e.g. the images of a cat with elephant texture. The Stylized-ImageNet dataset is designed to measure the model's ability to recognize shapes rather than textures, and higher performance in Table 3c is a direct proof of our model's efficacy in recognizing objects by shape. While ViT outperforms CNN on the task, our discrete representations yield another 10+% gain. However, if *the discrete token is replaced with the global, continuous CNN features*, such robustness gain is gone (cf. Table 8). This substantiates the benefit of discrete representations in recognizing object shapes.

Additionally, we conduct the "shape vs. texture biases" analysis following (Geirhos et al., 2019) under the OOD setting "IN-SIN" (cf. Appendix A.5). Figure. 5a compares shape bias between

Table 4: State-of-the-art mean Corruption Error (mCE) ↓ rate on ImageNet-C (the smaller the better).

| Model | mCE ↓ | Gauss | Shot | Impul | Defoc | Glass | Motion | Zoom | Snow | Frost | Fog | Bright | Contrast | Elastic | Pixel | JPEG |
|---|---|---|---|---|---|---|---|---|---|---|---|---|---|---|---|---|
| Resnet152 (He et al., 2016) | 69.27 | 72.5 | 73.4 | 76.3 | 66.9 | 81.4 | 65.7 | 74.5 | 70.7 | 67.8 | 62.1 | 51.0 | 67.1 | 75.6 | 68.9 | 65.1 |
| +Stylized (Geirhos et al., 2019) | 64.19 | 63.3 | 63.1 | 64.6 | 66.1 | 77.0 | 63.5 | 71.6 | 62.4 | 65.4 | 59.4 | 52.0 | 62.0 | 73.2 | 55.3 | 62.9 |
| +GenInt (Mao et al., 2021a) | 61.70 | 59.2 | 60.2 | 62.4 | 60.7 | 70.8 | 59.5 | 69.9 | 64.4 | 63.8 | 58.3 | 48.7 | 61.5 | 70.9 | 55.2 | 60.0 |
| DA (Hendrycks et al., 2021a) | 53.60 | - | - | - | - | - | - | - | - | - | - | - | - | - | - | - |
| ViT-B | 52.71 | 43.0 | 47.2 | 44.4 | 63.2 | 73.4 | 55.1 | 70.8 | 51.6 | 45.6 | 35.2 | 44.2 | 41.3 | 61.6 | 54.0 | 59.4 |
| ViT-B + Ours | **46.22** | 36.9 | 38.6 | 36.0 | 54.6 | 57.4 | 53.4 | 63.2 | 45.4 | 38.7 | 34.1 | 40.9 | 39.6 | 56.6 | 45.0 | 53.0 |
| ViT-B *21K | 49.62 | 47.0 | 48.3 | 46.0 | 55.7 | 65.6 | 46.7 | 54.0 | 40.0 | 40.6 | 32.2 | 42.3 | 42.0 | 57.1 | 63.1 | 63.6 |
| ViT-B + Ours *21K | **38.74** | 30.5 | 30.6 | 29.2 | 47.3 | 54.3 | 44.4 | 49.4 | 34.5 | 31.7 | 25.7 | 34.7 | 33.1 | 52.9 | 39.5 | 43.2 |

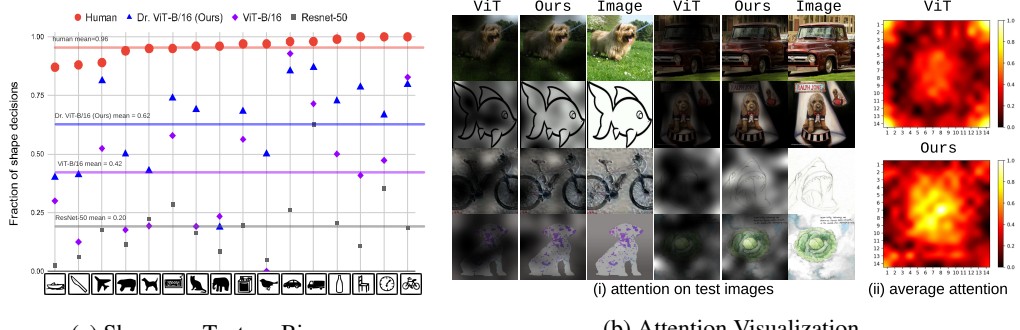

(a) Shape vs. Texture Biases.

(b) Attention Visualization.

Figure 5: We show the fraction of shape decisions on Stylized-ImageNet in Figure (a), and attention on OOD images in Figure (b), where (i) is the attention map, and (ii) is the heat map of averaged attention from images in a mini-batch. See Appendix A.1.4 for details.

humans and three models: ResNet-50, ViT-B, and Ours. It shows the scores on 16 categories along their average denoted by the colored horizontal line. Humans are highly biased towards shape with an average fraction of 0.96 to correctly recognize an image by shape. ViT (0.42) is more shape-biased than CNN ResNet-50 (0.20), which is consistent with the prior studies (Naseer et al., 2021). Adding discrete representation (0.62) greatly shrinks the gap between the ViT (0.42) and human baseline (0.96). Such behavior is not observed when adding global CNN features whose average fraction of shape decisions is 0.3, lower than the baseline ViT, hence is not displayed in the figure.

Finally, we validate discrete representation's ability in modeling shape information via position embedding. Following (Chen et al., 2021b), we compare training ViT with and without using position embedding. As position embedding is the only vehicle to equip ViT with shape information, its contribution suggests to what degree the model makes use of shape for recognition. As shown in Table 5, removing position embedding from the ViT model only leads to a marginal performance drop (2.8%) on ImageNet, which is consistent with (Chen et al., 2021b). However, without position embedding, our model accuracy drops by 29%, and degrades by a significant 36%-94% on the robustness benchmarks. This result shows that spatial information becomes crucial only when discrete representation is used, which also suggests our model relies on more shape information for recognition.

Table 5: Contribution of position embedding for robust recognition as measured by the relative performance drop when the position embedding is removed. Position embedding is much more crucial in our model.

| Model | ImageNet | Real | Rendition | Stylized | Sketch | ObjectNet | V2 | A | C ↓ |
|---|---|---|---|---|---|---|---|---|---|
| | | | Out of Distribution Robustness Test | | | | | | |
| ViT-B | 78.73 | 84.85 | 38.15 | 10.39 | 28.6 | 28.71 | 67.34 | 16.92 | 53.51 |
| -w/o. PosEmb | 76.51 | 77.01 | 28.25 | 5.86 | 15.17 | 24.02 | 63.22 | 13.13 | 69.99 |
| Relative drop (%) | 2.8% | 9.2% | 26.0% | 43.6% | 47.0% | 16.3% | 6.1% | 22.4% | 30.8% |
| Ours ViT-B | 79.48 | 84.86 | 44.77 | 19.38 | 34.59 | 30.55 | 68.05 | 17.20 | 46.22 |
| - w/o. PosEmb | 56.27 | 59.06 | 17.24 | 4.14 | 6.57 | 11.36 | 42.98 | 3.51 | 89.76 |
| Relative drop (%) | 29.2% | 30.4% | 61.5% | 78.6% | 81.0% | 62.8% | 36.8% | 79.6% | 94.2% |

**Qualitative results.** First, following (Dosovitskiy et al., 2020), we visualize the learned pixel embeddings, called filters, of the ViT-B model and compare them with ours in Fig. 3b with respect to *(1)* randomly selected filters (the top row in Fig. 3b) and *(2)* the first principle components (the bottom). We visualize the filters of our default model here and include more visualization in Appendix A.1.3. The evident visual differences suggest that our learned filters capture structural patterns.

Second, we compare the attention from the classification tokens in Fig. 5b, where *(i)* visualizes the individual images and *(ii)* averages attention values of all the image in the same mini-batch. Our model attends to image regions that are semantically relevant for classification and captures more global contexts relative to the central object.

Finally, we investigate what information is preserved in discrete representations. Specifically, we reconstruct images by the VQ-GAN decoder from the discrete tokens and finetuned discrete embeddings. Fig. 4 visualizes representative examples from ImageNet and the seven robustness benchmarks. By comparing the original and decoded images in Fig. 4, we find the discrete representation

preserves object shape but can perturb unimportant local signals like textures. For example, the decoded image changes the background but keeps the dog shape in the last image of the first row. Similarly, it hallucinates the text but keeps the bag shape in the first image of the third row. In the first image of ImageNet-C, the decoding deblurs the bird image by making the shape sharper. Note that the discrete representation is learned only on ImageNet, but it can generalize to the other out-of-distribution test datasets. We present more results with failure cases in Appendix A.1.1.

## 4.4 ABLATION STUDIES

**Model capacity vs. robustness.** Does our robustness come from using larger models? Fig. 6 shows the robustness vs. the number of parameters for the ViT baselines (blue) and our models (orange). We use ViT variants Ti, S, B, and two hybrid variants Hybrid-S and Hybrid-B, as baselines. We use "+Our" to denote our method and "+Small" to indicate that we use a smaller version of quantization encoder (cf. Appendix A.11.1). With a similar amount of model parameters, our model outperforms the ViT and Hybrid-ViT in robustness.

**Codebook size vs. robustness.** In Table 6, we conduct an ablation study on the size of the codebook, from 1024 to 8192, using the small variant of our quantized encoder trained on ImageNet. A larger codebook size gives the model more capacity, making the model closer to a continuous one. Overall, Stylized-ImageNet benefits the most when the feature is highly quantized. With a medium codebook size, the model strikes a good balance between quantization and capacity, achieving the best overall performance. A large codebook size learned on ImageNet can hurt performance.

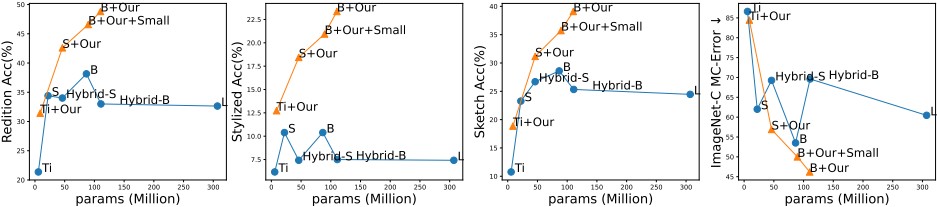

Figure 6: The robustness vs. #model-parameters on 4 robust test set. Our models (orange) achieve better robustness with a similar model capacity.

Table 6: The impact of codebook size for robustness.

| CodeBook Size | ImageNet | Real | Rendition | Stylized | Sketch | ObjectNet | V2 | A | C ↓ |
|---|---|---|---|---|---|---|---|---|---|
| | | | | Out of Distribution Robustness Test | | | | | |
| 1024 | 76.63 | 77.69 | 45.92 | **21.95** | 35.02 | 26.03 | 64.18 | 10.68 | 58.64 |
| 4096 | **77.31** | **78.17** | **47.04** | 21.33 | 35.71 | **27.72** | **64.79** | **11.37** | **57.26** |
| 8192 | 77.04 | 77.92 | 46.58 | 20.94 | **35.72** | 27.54 | 65.23 | 11.00 | 57.89 |

**Discrete vs. Continuous global representation.** In Appendix A.4, instead of using our discrete token, we study whether concatenate continuous CNN representations with global information can improve robustness. Results show that concatenating global CNN representation performs no better than the baseline, which demonstrates the necessity of discrete representations.

**Network designs for combining the embeddings.** In Appendix A.9, we experiment with 4 different designs, including addition, concatenation, residual gating (Vo et al., 2019), and cross-attention, to combine the discrete and pixel embeddings. Among them, the simple concatenation yields the best overall result.

## 5 CONCLUSION

This paper introduces a simple yet highly effective input representations for vision transformers, in which an image patch is represented as the combined embeddings of pixel and discrete tokens. The results show the proposed method is generic and works with several vision transformer architectures, improving robustness across seven out-of-distribution ImageNet benchmarks. Our new findings connect the robustness of vision transformer to discrete representation, which hints towards a new direction for understanding and improving robustness as well as a joint vision transformer for both image classification and generation.

## 6 ACKNOWLEDGEMENTS

We thank the discussions and feedback from Han Zhang, Hao Wang, and Ben Poole.

**Ethics statement:** The authors attest that they have reviewed the ICLR Code of Ethics for the 2022 conference and acknowledge that this code applies to our submission. Our explicit intent with this research paper is to improve the state-of-the-art in terms of accuracy and robustness for Vision Transformers. Our work uses well-established datasets and benchmarks and undertakes detailed evaluation of these with our novel and improved approaches to showcase state-of-the-art improvements. We did not undertake any subject studies or conduct any data-collection for this project. We are committed in our work to abide by the eight General Ethical Principles listed at ICLR Code of Ethics (`https://iclr.cc/public/CodeOfEthics`).

**Reproducibility statement** Our approach is simple in terms of implementation, as we only change the input embedding layer for the standard ViT while keeping everything else, such as the data augmentation, the same as the established ViT work. We use the standard, public available training and testing dataset for all our experiments. We include flow graph for our architecture in Fig. 1, pseudo JAX code in Fig. 3a, and implementation details in Section 4.1 and Appendix A.11. We will release our model and code to assist in comparisons and to support other researchers in reproducing our experiments and results.

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

# A  APPENDIX

The Appendix is organized as follows. Appendix A.2 extends the discussion on learning objective. Appendix A.1 presents more figures to visualize the attention, the decoded images, and the decoding failure cases. Appendix A.9 compares designs for combing the pixel and discrete embeddings. In the end, Appendix A.11 discusses the implementation details.

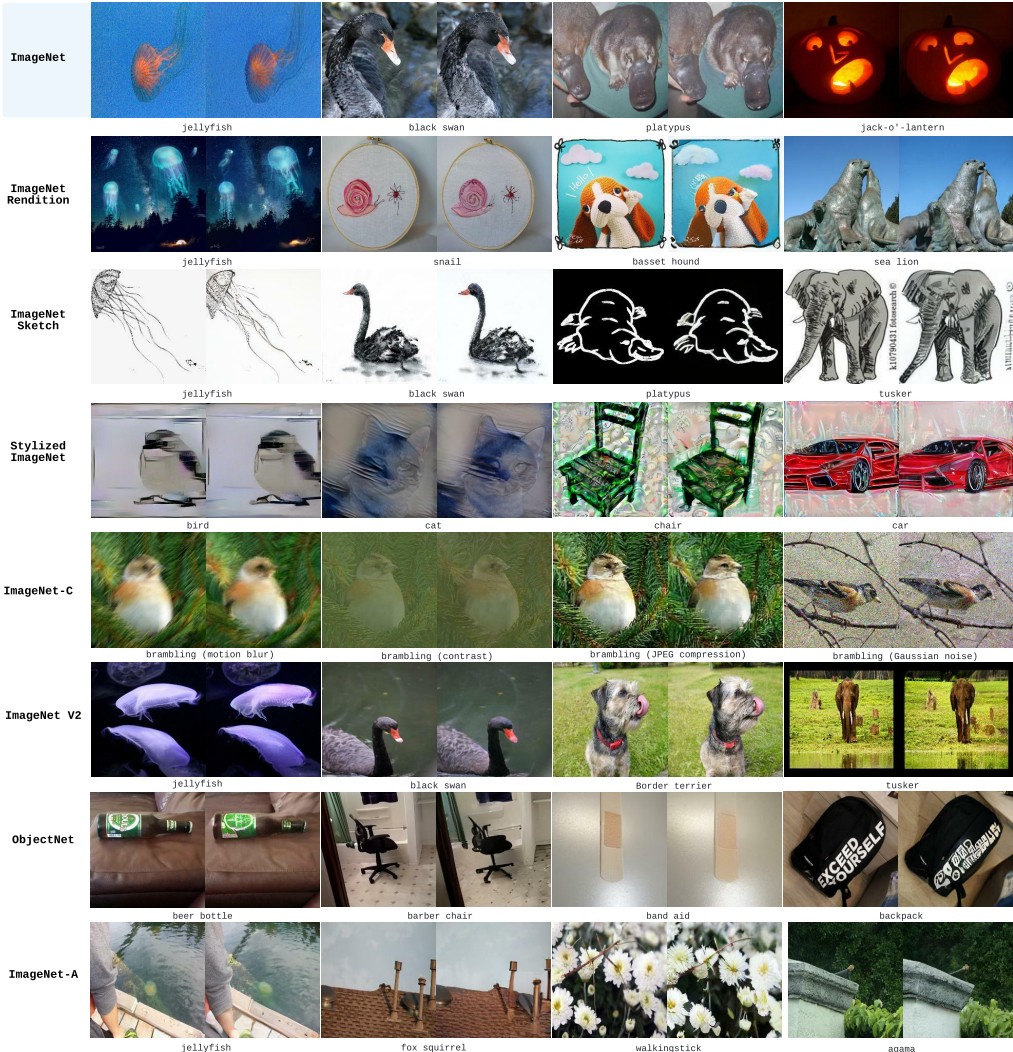

Figure 7: Visualization of eight evaluation benchmarks. Each image consists of the original test image (Left) and the decoded image (Right) from the finetuned discrete embeddings. The encoder and decoder are trained only on ImageNet 2012 data but generalize on out-of-distribution datasets.

## A.1  VISUALIZATION

### A.1.1  RECONSTRUCTED IMAGES FROM DISCRETE REPRESENTATION

Fig. 7 shows more examples of reconstructed images decoded from the finetuned discrete embedding. Generally, the discrete representation reasonably preserves object shape but can perturb local signals. Besides, Fig 7 also shows the distribution diversity in the experimented robustness benchmarks, e.g., the objects in ImageNet-A are much smaller. Note that while the discrete representation is learned only on the ImageNet 2012 training dataset (the first row in Fig. 7), it can generalize to the other out-of-distribution test datasets.

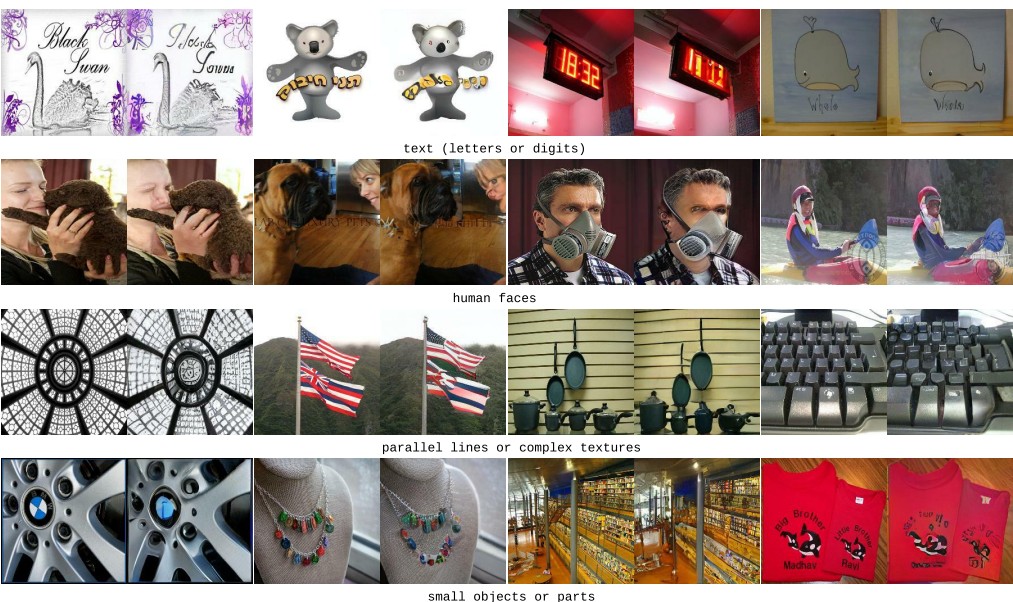

Figure 8: Visualization of failure cases for the decoded images. Each image consists of the original test image (Left) and the decoded image (Right) from the finetuned discrete embeddings.

### A.1.2 LIMITATIONS OF DISCRETE REPRESENTATION

We find four failure cases that the discrete representation are unable to capture: text, human faces, parallel lines, and small objects or parts. The example images are illustrated in Fig. 8. Without prior knowledge, the discrete representation has difficulty to reconstruct text in image. On the other hand, it is interesting to find the decoder can model animal faces but not human faces. This may be a result of lacking facial images in the training data of ImageNet and ImageNet-21K.

The serious problem we found for recognition is its failure to capture small objects or parts. Though this can be beneficial sometimes, e.g. force the model to recognize the tire without using the "BMW" logo (see the first image in the last row of Fig. 8), it can cause problems in many cases, e.g. recognizing the small whales in the last image. As a result, our proposed model leverages the power of both embeddings to promote the interaction between modeling global and local features.

### A.1.3 FILTERS

In Fig. 3b of the main paper, we illustrate the visual comparison of the learned pixel embeddings between the standard ViT and our best-performing model. In this subsection, we extend the visualization to our models with a varying number of pixel dimensions when concatenating the discrete and pixel embedding, where their classification performances are compared in Table 13. As the dimension of pixel embedding grows, the filters started to pick up more high-frequency signals. However, the default setting in the paper using only 32 pixel dimensions (pixel_dim=32), which seems to induce an inductive bias by limiting the budget of pixel representation, turns out to be the best performing model.

### A.1.4 ATTENTIONS

We visualize the attention following the steps in (Dosovitskiy et al., 2020). To be specific, to compute maps of the attention from the output token to the input space, we use Attention Rollout (Abnar & Zuidema, 2020). Briefly, we average attention weights of ViT-B and ViT-B+Ours across all heads and then recursively multiply the weight matrices of all layers. Both ViT and ViT+Ours are trained on ImageNet.

In Fig. 11, Fig. 12 and Fig. 13, we visualize the attention of individual images in the first mini-batch of each test set. As shown, our model attends to image regions that are semantically relevant for

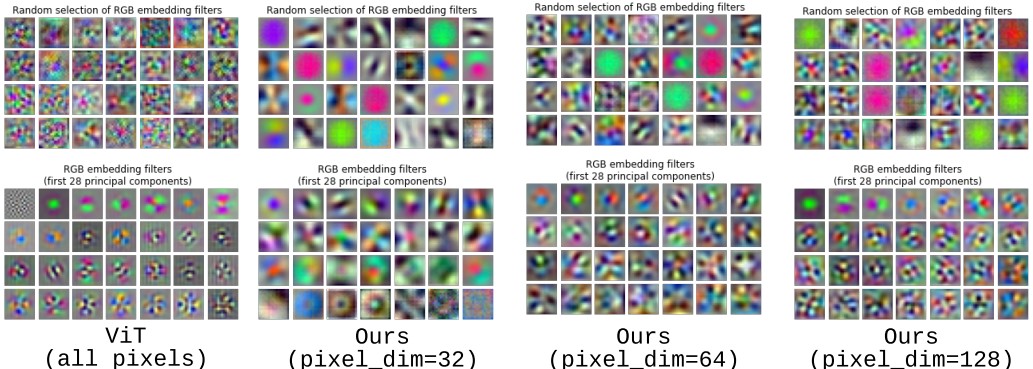

Figure 9: Comparing visualized pixel embeddings of the ViT and our model. The top row shows the randomly selected filters and the Bottom shows the first 28 principal components. Our models with varying pixel dimensions are shown, where their classification performances are compared in Table 13. Ours (pixel_dim=32) works the best and is used as the default model in the main paper.

classification and captures more global contexts relative to the central object. This can be better seen from Fig. 10, where the heat map averages the attention values of all images in the first mini-batch of each test set. As found in prior works (Dosovitskiy et al., 2020; Naseer et al., 2021), ViTs put much attentions on the corners of the image, our attentions are more global and do not exhibit the same defect.

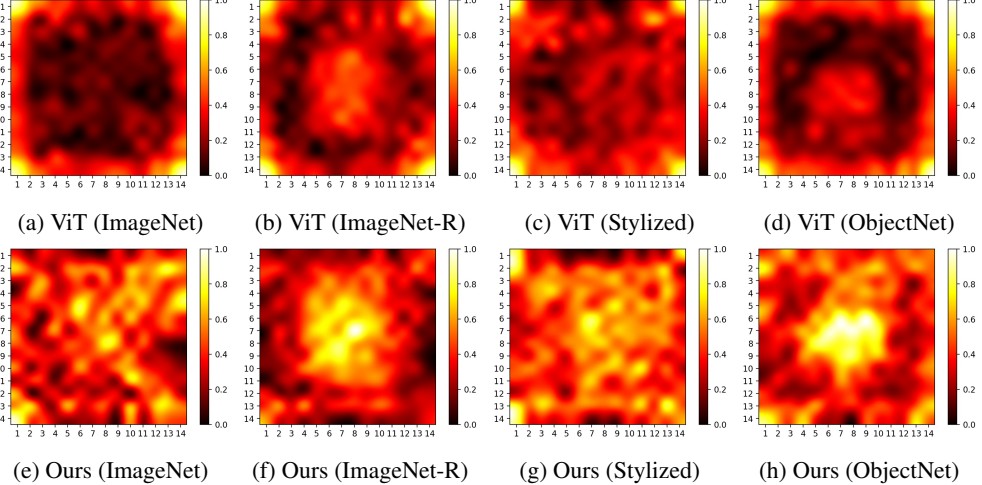

Figure 10: Comparison of average attention of the ViT (top row) and the proposed model (bottom row) on four validation datasets: ImageNet 2012, ImageNet-R, Stylized-ImageNet, and ObjectNet. The heat map averages the attention values of all images in the first mini-batch of each test set. The results show our attention capture more global context relative to the central object.

## A.2 DETAILS ON LEARNING OBJECTIVE

Let $x$ denote an image with the class label $y$, and $z$ represent the discrete tokens $z \in \mathbb{Z}$. For notational convenience, we use a single random variable to represent the discrete latent variables whereas our implementation actually represents an image as a flattened 1-D sequence following the raster scan.

- $q_\phi(z|x)$ denotes the VQ-VAE encoder, parameterized by $\phi$, to represent an image $x$ into the discrete token $z$.

- $p_\theta(x|z)$ is the decoder that models the distribution over the RGB image generated from discrete tokens.
- $p_\psi(y|x, z)$ stands for the vision transformer model shown in Fig. 1 of the main paper.

We model the joint distribution of image $x$, label $y$, and the discrete token $z$ using the factorization $p(x, y, z) = p_\theta(x|z)p_\psi(y|x, z)p(z)$. Our overall procedure can be viewed as maximizing the evidence lower bound (ELBO) on the joint likelihood, which yields:

$$D_{\mathrm{KL}}[q_\phi(z|x), p(z|x, y))] = -\sum_z q_\phi(z|x) \log \frac{p(z|x, y)}{q_\phi(z|x)} \tag{9}$$

$$= -\sum_z q_\phi(z|x) \log \frac{p(x, y, z)}{p(x, y)q_\phi(z|x)} \tag{10}$$

Since the $f$-divergence is non-negative, we have:

$$-\sum_z q_\phi(z|x) \log \frac{p(x, y, z)}{q_\phi(z|x)} + \log p(x, y)[\sum_z q_\phi(z|x)] \geq 0 \tag{11}$$

$$\log p(x, y) \geq \sum_z q_\phi(z|x) \log \frac{p(x, y, z)}{q_\phi(z|x)} \tag{12}$$

Using the factorization, we have:

$$\log p(x, y) \geq \mathbb{E}_{q_\phi(z|x)}[\log \frac{p_\psi(y|x, z)p_\theta(x|z)p(z)}{q_\phi(z|x)}] \tag{13}$$

Equation 13 yields the ELBO:

$$\log p(x, y) \geq \mathbb{E}_{q_\phi(z|x)}[\log p_\theta(x|z)] + \mathbb{E}_{q_\phi(z|x)}[\log \frac{p_\psi(y|x, z)p(z)}{q_\phi(z|x)}] \tag{14}$$

$$\geq \mathbb{E}_{q_\phi(z|x)}[\log p_\theta(x|z)] - D_{\mathrm{KL}}[q_\phi(z|x)\|p_\psi(y|x, z)p(z)] \tag{15}$$

In the first stage, we maximize the ELBO with respect to $\phi$ and $\theta$, which corresponds to learning the VQ-VAE encoder and decoder. We assume a uniform prior for both $p_\psi(y|x, z)$ and $p(z)$. Given that $q_\phi(z|x)$ predicts a one-hot output, the regularization term (KL divergence) will be a constant $\log K$, where $K$ is the size of the codebook $\mathbf{V}$. As in the VQ-VAE model, the distribution $p_\theta$ is deterministic. We note a similar assumption as in the DALL-E model (Ramesh et al., 2021) is used to stabilize the training, which means the transformer is not learned at this stage. We have the same observation as in DALL-E that this strategy works better than jointly training with $p_\psi(y|x, z)$.

In addition to the reconstruction loss, the VQ-VAE's training objective adds a dictionary learning loss and a commitment loss, calculated from:

$$\mathcal{L}_{\text{VQ-VAE}} = \log p(x|z_q(x)) + \|\text{sg}[z_e(x)] - \boldsymbol{v}\|^2 + \beta\|z_e(x) - \text{sg}[\boldsymbol{v}]\|, \tag{16}$$

where $z_e(x)$ and $z_q(x)$ represent the encoder output and the decoder input, respectively. $\boldsymbol{v}$ denotes the discrete embedding for the image $x$. $\text{sg}(\cdot)$ is the stop gradient function. The training used straight-through estimation which just copies gradients from $z_q(x)$ to $z_e(x)$. Notice that our implementation uses VQ-GAN (Esser et al., 2021) which adds an additional GAN loss:

$$\mathcal{L}_{\text{VQ-GAN}} = \log p(x|z_q(x)) + \|\text{sg}[z_e(x)] - \boldsymbol{v}\|^2 + \beta\|z_e(x) - \text{sg}[\boldsymbol{v}]\| + \lambda\mathcal{L}_{\text{GAN}}, \tag{17}$$

where $\beta = 0.25$ and $\lambda = 0.1$ are used to balance different loss terms. $\mathcal{L}_{\text{VQ-GAN}}$ also has a perceptual loss (Johnson et al., 2016).

In the second stage of finetuning, we optimize $\psi$ while holding $\phi$ and $\theta$ fixed, which corresponds to learning a vision transformer for classification and finetuning the discrete embedding $\mathbf{V}$. By rearranging Equation 14, we have:

$$\log p(x, y) \geq \mathbb{E}_{q_\phi(z|x)}[\log p_\psi(y|x, z)] + \mathbb{E}_{q_\phi(z|x)}[\log \frac{p_\theta(x|z)p(z)}{q_\phi(z|x)}], \tag{18}$$

where $p_\psi(y|x, z)$ is the proposed ViT model as illustrated in Fig. 1 of the main paper. The first term denotes the likelihood for classification that is learned by minimizing the multi-class cross-entropy loss. The second term becomes a constant given the fixed $\theta$ and $\phi$.

In the finetuning stage, two sets of parameters are updated: the vision transformer $\psi$ is learned from scratch and the discrete embedding $\mathbf{V}$ is finetuned. We fix the encoder and stop the gradient propagating back to the encoder considering the instability of straight-through estimation and the stop gradient operation in the dictionary learning loss, i.e. Equation 16. Empirically, we also compared the proposed pretraining-finetuning strategy with the joint learning strategy in which all parameters are optimized simultaneously. We find the joint learning significantly underperforms the proposed learning strategy.

## A.3 COMPARING TO DATA AUGMENTATIONS

Since our model only changes the token representation in the ViT architecture, it is conceptually different and complementary to data augmentation. Nevertheless, this subsection compares our method with seven combinations of recent data augmentation strategies, including CutOut (DeVries & Taylor, 2017), CutMix (Yun et al., 2019), RandAug (Cubuk et al., 2020), and Mixup (Zhang et al., 2018). As discussed following (Steiner et al., 2021), the ViT baseline and our model both use the default augmentation (RandAug + Mixup).

As shown in Table 7, our method achieves the best performance on the robustness benchmarks while is marginally worse than the best baseline with state-of-the-art data augmentation strategy (CutMix + RandAug) on the ImageNet in-distribution validation set. Incorporating state-of-the-art data augmentation in our model is worthy of future research.

Table 7: Model performance trained on ImageNet. We compared with several recent data augmentations, including CutOut (DeVries & Taylor, 2017) and CutMix Yun et al. (2019), on the ViT-B model. All columns indicate the top-1 classification accuracy except the last column ImageNet-C which indicates the mean Corruption Error (lower the better). The bold number indicates higher accuracy than the corresponding baseline. The box highlights the best accuracy. While the state-of-the-art data augmentation (CutMix+RandAug) can improve Imagenet in-distribution accuracy, they are worse than ours on the OOD test accuracy.

| Model | ImageNet | Real | Rendition | Stylized | Sketch | ObjectNet | V2 | A | C ↓ |
|---|---|---|---|---|---|---|---|---|---|
| | | | Out of Distribution Robustness Test | | | | | | |
| ViT-B No Augmentation | 72.47 | 78.69 | 24.56 | 5.94 | 14.34 | 13.44 | 59.32 | 5.23 | 88.72 |
| + Ours (discrete only) | **73.73** | **79.63** | **34.61** | **8.94** | **23.66** | **20.84** | **60.30** | **6.45** | **74.82** |
| ViT-B + CutOut | 72.27 | 77.91 | 25.74 | 7.89 | 16.50 | 17.82 | 58.05 | 10.23 | 69.01 |
| ViT-B + CutMix | 75.51 | 80.53 | 28.45 | 7.89 | 17.15 | 21.62 | 62.36 | 14.72 | 64.07 |
| ViT-B + CutOut + Mixup | 71.31 | 77.07 | 25.07 | 6.17 | 15.55 | 16.28 | 56.84 | 8.33 | 76.44 |
| ViT-B + CutMix + Mixup | 71.66 | 76.82 | 23.46 | 4.92 | 13.91 | 16.99 | 56.98 | 10.23 | 80.54 |
| ViT-B + CutOut + RandAug | 79.07 | 84.64 | 38.10 | 12.34 | 26.92 | 28.32 | 66.88 | 15.77 | 54.52 |
| ViT-B + CutMix + RandAug | **79.63** | **85.24** | 38.34 | 10.94 | 27.46 | 29.80 | 68.05 | **18.80** | 51.63 |
| ViT-B + RandAug + Mixup | 78.73 | 84.85 | 38.15 | 10.39 | 28.60 | 28.71 | 67.34 | 16.92 | 53.51 |
| Our ViT-B + RandAug + Mixup | 79.48 | 84.86 | **44.77** | **19.38** | **34.59** | **30.55** | **68.05** | 17.20 | **46.22** |

## A.4 THE IMPORTANCE OF DISCRETE INFORMATION

This subsection compares to a new baseline that directly concatenates the global feature extracted by CNN to the input token of ViT. The only difference between this baseline and our model is that ours concatenates discrete embeddings rather than global CNN features. In Table 8, we extensively search the concatenation dimension of the CNN features in [64, 128, 256, 384, 512, 640]. However, none of the models improve robustness. Our results show that concatenating the global CNN feature performs no better than the ViT baseline and significantly worse than concatenating discrete representations, demonstrating the necessity of discrete representations for robustness.

## A.5 TEXTURE VS. SHAPE STUDY

In Fig. 5a of the main paper, we performs the "shape vs. texture biases" analysis following (Geirhos et al., 2019) (see Figure 4 in their paper). It uses the score "fraction of shape decision" to quantify a

Table 8: We replace our discrete token representation with global continuous (GC) features from the CNN model that has the same CNN architecture as our VQGAN encoder. We denote the dimension for the global features as GF-Dim. We vary the dimension of the pixel token to concatenate with the global continuous CNN features. Simply concatenating the global feature extracted from CNN does not improve robustness.

| Model | GF-Dim | ImageNet | Real | Out of Distribution Robustness Test | | | | | | |
| | | | | Rendition | Stylized | Sketch | ObjectNet | V2 | A | C ↓ |
|---|---|---|---|---|---|---|---|---|---|---|
| GC CNN + ViT | 640 | 78.48 | 83.27 | 34.75 | 9.38 | 24.85 | 28.37 | 65.65 | 16.13 | 58.71 |
| GC CNN + ViT | 512 | 78.59 | 83.24 | 34.31 | 10.23 | 25.50 | 27.50 | 65.35 | 16.48 | 57.66 |
| GC CNN + ViT | 384 | 78.51 | 83.44 | 34.66 | 10.47 | 25.14 | 28.35 | 65.86 | 15.97 | 57.64 |
| GC CNN + ViT | 256 | 78.44 | 83.09 | 34.96 | 9.61 | 25.20 | 27.66 | 65.95 | 15.96 | 58.35 |
| GC CNN + ViT | 128 | 78.29 | 82.96 | 34.32 | 9.22 | 24.90 | 26.91 | 65.28 | 15.73 | 58.71 |
| GC CNN + ViT | 64 | 78.34 | 82.98 | 34.35 | 9.19 | 25.06 | 27.07 | 65.48 | 15.40 | 58.58 |
| ViT | 0 | 78.73 | 84.85 | 38.15 | 10.39 | 28.60 | 28.71 | 67.34 | 16.92 | 53.51 |
| Ours (discrete only) | - | 78.67 | 84.28 | **48.82** | **22.19** | **39.10** | 30.27 | 66.52 | 14.77 | 55.21 |
| Ours | - | 79.48 | 84.86 | **44.77** | **19.38** | **34.59** | 30.55 | 68.05 | 17.20 | **46.22** |

model's shape-bias between 0 and 1. To compute the score, we follow the steps on their Github[1]as follows: 1) evaluate the model on all 1280 images in Stylized ImageNet; 2) map the model decision to 16 classes; 3) exclude images without a cue conflict; 4) take the subset of "correctly" classified images (either shape or texture category correctly predicted); (5) compute "shape bias" as the fraction between correct shape decisions and (correct shape decisions + correct texture decisions).

Fig. 5a in the main paper presents the scores on 16 categories along their average denoted by the colored horizontal line. We compute the scores for three models trained on ImageNet, i.e. ResNet-50, ViT-B/16, and ours, and quote human scores from (Geirhos et al., 2019). We also compute the score for the concatenating global CNN features baseline which is about 0.3. Note that the OOD generalization setting "IN-SIN" (Geirhos et al., 2019) is used, where the model is trained only on the ImageNet (IN) and tested on the Stylized ImageNet (SIN) images with conflicting shape and texture cue.

## A.6 THE EFFECT OF CODEBOOK ENCODER CAPACITY ON ROBUSTNESS

Table 9: The impact of codebook encoder capacity for robustness.

| CodeBook Size | ImageNet | Real | Out of Distribution Robustness Test | | | | | | |
| | | | Rendition | Stylized | Sketch | ObjectNet | V2 | A | C ↓ |
|---|---|---|---|---|---|---|---|---|---|
| Small VQ Encoder | 76.63 | 77.69 | 45.92 | 21.95 | 35.02 | 26.03 | 64.18 | 10.68 | 58.64 |
| VQ Encoder | **78.67** | **84.28** | **48.82** | **22.19** | **39.10** | **30.27** | **66.52** | **14.77** | **55.21** |

We analyze how the capacity of the discrete encoder affects the model's robustness. We train a small encoder and a standard encoder and compare their performance. We describe the VQ encoder's architecture configuration in Table 17, where the small encoder has only 13% of the #parameters of the standard VQ encoder. We run experiments in Table 9. Our results show that using the standard VQ-encoder is better but the small VQ-encoder comparatively yields reasonable results considering the reduced model capacity.

## A.7 THE EFFECT OF PRETRAINING CODEBOOK WITH MORE DATA ON ROBUSNTESS

Table 10: The impact of pretraining codebook on sufficiently large data for robustness. The codebook is pretrained on the standard ImageNet ILSVRC 2012 or the ImageNet21K dataset (about 11x larger). Using the codebook, we train ViT models on ImageNet21K.

| Codebook Pretrained Data | ImageNet | Real | Out of Distribution Robustness Test | | | | | | |
| | | | Rendition | Stylized | Sketch | ObjectNet | V2 | A | C ↓ |
|---|---|---|---|---|---|---|---|---|---|
| ImageNet | 83.32 | 88.36 | 51.56 | 14.77 | 41.10 | 42.50 | 72.57 | 33.51 | 55.31 |
| ImageNet21K | **83.40** | **88.44** | **55.26** | **19.69** | **44.72** | **43.92** | **72.68** | **36.64** | **45.86** |

We analyze the effect of pre-training codebook on sufficiently large data. We use the same model architecture, but pretrain two codebooks (and encoders) on the ImageNet ILSVRC 2012 and the

---

[1]https://github.com/rgeirhos/texture-vs-shape

ImageNet21K (about 11x larger), respectively. Using the pre-trained codebook, we train ViTs on ImageNet21K and compare the results in Table 10, where it shows pretraining on large data improves the robustness.

## A.8 ADDITIONAL MODELS FOR LEARNING DISCRETE REPRESENTATION

The results show that the ability of discrete representations to improve ViT's robustness is general which is not limited to specific VQ-GAN models.

Table 11: Model architecture for discrete representation. We **bold** the numbers if they improve robustness over the baseline ViT model. The numbers are boxed where VQ-VAE further improves robustness than VQ-GAN under apple-to-apple comparison.

| Codebook Pretrained Model | ImageNet | Real | Rendition | Stylized | Sketch | ObjectNet | V2 | A | C ↓ |
|---|---|---|---|---|---|---|---|---|---|
| | | | | Out of Distribution Robustness Test | | | | | |
| Baseline (No discrete) | 78.73 | 84.85 | 38.15 | 10.39 | 28.60 | 28.71 | 67.34 | 16.92 | 53.51 |
| VQ-VAE (Discrete Only) | 78.36 | 84.35 | **46.22** | **23.36** | **35.17** | **29.34** | 66.24 | 13.61 | **52.20** |
| VQ-GAN (Discrete Only) | 78.67 | 84.28 | **48.82** | **22.19** | **39.10** | **30.27** | 66.52 | 14.77 | 55.21 |
| VQ-VAE | 78.51 | 83.68 | **41.54** | **17.50** | **30.91** | 27.43 | 65.74 | 15.61 | **50.47** |
| VQ-GAN | **79.48** | **84.86** | **44.77** | **19.38** | **34.59** | **30.55** | **68.05** | **17.20** | **46.22** |

## A.9 COMPARING DESIGNS FOR COMBINING EMBEDDINGS

In this subsection, we compare different designs to combine the pixel and discrete embeddings. See the combination operation in Fig. 1 and in Equation 6 of the main paper. The results are used verify our design of using a simple concatenation presented in the paper. Four designs are considered and discussed below. Their performances are compared in Table 12.

Table 12: The accuracy using different combination methods. While directly adding pixel token to discrete token improves the most ImageNet and ImageNet-A accuracy, concatenation method improves the overall robustness.

| Combining Method | ImageNet | Real | Rendition | Stylized | Sketch | ObjectNet | V2 | A | C ↓ |
|---|---|---|---|---|---|---|---|---|---|
| | | | | Out of Distribution Robustness Test | | | | | |
| Addition | **79.67** | 84.89 | 40.68 | 13.59 | 30.89 | 29.18 | 67.19 | **17.68** | 50.85 |
| Concatenation | 79.48 | **84.86** | **44.77** | **19.38** | **34.59** | **30.55** | **68.05** | 17.20 | **46.22** |
| Residual Gating | 74.59 | 79.66 | 41.06 | 17.03 | 29.56 | 25.34 | 62.37 | 9.33 | 59.88 |
| Cross-Attention | 79.18 | 84.53 | 43.75 | 18.67 | 34.19 | 29.71 | 66.73 | 15.81 | 50.06 |

**Addition.** We first use linear projection matrix $\mathbf{E}$ to obtain the pixel embedding that shares the same dimension as the discrete embedding which is 256. We then add the two embedding, and project the resulting embedding to the dimension required by the ViT if needed. Formally, we compute

$$f(\mathbf{V}_{\mathbf{z}_d}, \mathbf{x}_p\mathbf{E}) = (\mathbf{V}_{\mathbf{z}_d} + \mathbf{x}_p\mathbf{E})\mathbf{E}_2, \tag{19}$$

where $\mathbf{V}_{\mathbf{z}_d}$ and $\mathbf{x}_p\mathbf{E}$ represent the discrete and pixel embeddings, respectively, and $\mathbf{E}_2$ is a new MLP projection layer that is needed when the resulting dimension is different to the ViT input.

**Concatenation.** We concatenate the discrete embedding with the pixel embedding, and then feed the resulting embedding into the transformer encoder. By default, we use a dimension of 32 for the pixel embedding, a dimension of 256 for the discrete embedding, thus we pad 0 to the vector if the input dimension of the ViT is higher than 288.

$$f(\mathbf{V}_{\mathbf{z}_d}, \mathbf{x}_p\mathbf{E}) = [\mathbf{V}_{\mathbf{z}_d}; \mathbf{x}_p\mathbf{E}; \mathbf{0}], \tag{20}$$

where ";" indicates the concatenation operation.

**Residual Gating.** As pixel embeddings may contain nuisances, inspired by (Vo et al., 2019), we learn a gate from both embeddings, and then multiply it with the pixel embeddings to filter out details that are unimportant to classification. Specifically, we calculate the gate by a 2-layer MLP, and apply a softmax to the output from the last layer.

$$\mathbf{G} = \text{Softmax}(\text{MLP}(\mathbf{V}_{\mathbf{z}_d}; \mathbf{x}_p \mathbf{E})) \tag{21}$$

Then the pixel embeddings are gated by:

$$\mathbf{P} = \mathbf{G} \odot \mathbf{x}_p \mathbf{E} \tag{22}$$

Then we concatenate the gated embedding with the discrete embedding:

$$f(\mathbf{V}_{\mathbf{z}_d}, \mathbf{x}_p \mathbf{E}) = [\mathbf{V}_{\mathbf{z}_d}; \mathbf{P}; \mathbf{0}] \tag{23}$$

**Cross-Attention.** We use the discrete embedding as the query, and the pixel token as the key and value in a standard multi-head cross-attention module (MCA).

$$\mathbf{A} = \text{MCA}(\mathbf{V}_{\mathbf{z}_d}, \mathbf{x}_p \mathbf{E}, \mathbf{x}_p \mathbf{E}) \tag{24}$$

We then concatenate the attended feature output with the discrete embedding. We also pad $\mathbf{0}$ if the ViT requires larger input dimensionality.

$$f(\mathbf{V}_{\mathbf{z}_d}, \mathbf{x}_p \mathbf{E}) = [\mathbf{V}_{\mathbf{z}_d}; \mathbf{A}; \mathbf{0}] \tag{25}$$

### A.9.1 THE ROBUSTNESS EFFECT OF USING CONTINUOUS REPRESENTATION WITH DISCRETE REPRESENTTION.

The above study shows that concatenation is the most effective way to combine the discrete and continuous representations. As we fixed the dimensionality for the discrete token to be 256, we can study the optimal ratio between continuous and discrete representations by changing the dimensionality on the pixel token.

Table 13: The robust performance under different pixel token dimension in concatenation combining method. By increasing the dimension of the pixel embeddings, the model uses more continuous representations than discrete representations. There is an inherent trade off between the out of distribution robustness performance. The more continuous representations the model use, the lower robustness on the out-of-distribution set that depicting object shape, but also aciheves higher accuracy on the in-distribution ImageNet and out of distribution variants ImageNet-A that requires detailed information. However, using only continuous embeddings also hurt robustness. Thus we choice to use dimension 32, which balance the trade-off and achieves the best overall robustness.

| Pixel Token Dimension | ImageNet | Real | Rendition | Stylized | Sketch | ObjectNet | V2 | A | C ↓ |
|---|---|---|---|---|---|---|---|---|---|
| | | | Out of Distribution Robustness Test | | | | | | |
| 0 | 78.67 | 84.28 | 48.82 | 22.19 | 39.10 | 30.27 | 66.52 | 14.77 | 55.21 |
| 8 | 79.05 | 84.50 | 46.08 | 21.17 | 34.67 | 30.20 | 66.71 | 15.15 | 48.42 |
| 16 | 79.45 | 84.84 | 46.00 | 20.47 | 34.88 | 29.42 | 67.09 | 15.81 | 46.91 |
| 32 | 79.48 | 84.86 | 44.77 | 19.38 | 34.59 | 30.55 | 68.05 | 17.20 | 46.22 |
| 64 | 79.71 | 84.98 | 43.97 | 18.75 | 33.89 | 29.91 | 67.98 | 17.47 | 46.10 |
| 128 | 80.04 | 85.07 | 41.96 | 15.47 | 31.65 | 30.05 | 68.03 | 17.80 | 49.58 |
| All Pixel | 78.73 | 84.85 | 38.15 | 10.39 | 28.60 | 28.71 | 67.34 | 16.92 | 53.51 |

### A.10 THE IMPORTANCE OF SPATIAL INFORMATION FOR DIFFERNT ARCHITECTURES

In addition to the analysis in the main paper, we also investigate whether it is the convolutional operation in the vector quantized encoder that makes the model use the spatial information better. We remove the position embedding from the Hybrid-B model, whose input is also produced by a convolution encoder. Our results show that removing the positional information from the Hybrid model with a convolutional encoder does not decrease the performance. However, for both our models, the discrete only and the combined one, removing positional embedding causes a large drop in performance. This shows that our robustness gain via the spatial structure is from our discrete design, not convolution.

Table 14: Contribution of position embedding for robust recognition as measured by the relative performance drop when the position embedding is removed. We experiment on ViT, Ours, Hybrid-ViT, and Ours with discrete token only. Note that Hybrid uses the continuous, global feature from a ResNet-50 CNN as the input token. A larger relative drop indicates the model relies on spatial information more. Adding discrete token input representation drives position information and spatial structure more crucial in our model.

| Model | ImageNet | Real | Out of Distribution Robustness Test | | | | | | |
| | | | Rendition | Stylized | Sketch | ObjectNet | V2 | A | C↓ |
|---|---|---|---|---|---|---|---|---|---|
| ViT-B | 78.73 | 84.85 | 38.15 | 10.39 | 28.6 | 28.71 | 67.34 | 16.92 | 53.51 |
| -w/o. PosEmb | 76.51 | 77.01 | 28.25 | 5.86 | 15.17 | 24.02 | 63.22 | 13.13 | 69.99 |
| Relative drop (%) | 2.8% | 9.2% | 26.0% | 43.6% | 47.0% | 16.3% | 6.1% | 22.4% | 30.8% |
| Hybrid-ViT-B | 74.94 | 80.54 | 33.03 | 7.5 | 25.33 | 23.08 | 61.30 | 7.44 | 69.61 |
| Hybrid-ViT-B- w/o. PosEmb | 75.13 | 80.66 | 33.11 | 7.5 | 25.15 | 22.78 | 61.75 | 7.60 | 68.51 |
| relative drop (%) | -0.3 | -0.1 | -0.2 | 0 | 0.7 | 3.0 | 0.7 | 2.1 | 1.6 |
| Ours (discrete only) | 78.67 | 84.28 | 48.82 | 22.19 | 39.10 | 30.27 | 66.52 | 14.77 | 55.21 |
| Ours (discrete only) w/o. PosEmb | 16.34 | 5.09 | 1.95 | 2.09 | 1.20 | 1.75 | 11.75 | 1.29 | 124 |
| Relative Drop (%) | 79.2 | 94.1 | 96.0 | 90.5 | 96.9 | 94.2 | 82.3 | 91.3 | 124.6 |
| Ours | 79.48 | 84.86 | 44.77 | 19.38 | 34.59 | 30.55 | 68.05 | 17.20 | 46.22 |
| Ours- w/o. PosEmb | 56.27 | 59.06 | 17.24 | 4.14 | 6.57 | 11.36 | 42.98 | 3.51 | 89.76 |
| Relative drop (%) | 29.2% | 30.4% | 61.5% | 78.6% | 81.0% | 62.8% | 36.8% | 79.6% | 94.2% |

Table 15: Model configuration for the Transformer Encoder.

| Model | Layers | Hidden size | MLP size | Heads | #Params |
|---|---|---|---|---|---|
| Tiny (Ti) | 12 | 192 | 768 | 3 | 5.8M |
| Small (S) | 12 | 384 | 1546 | 6 | 22.2M |
| Base (B) | 12 | 768 | 3072 | 12 | 86M |
| Large (L) | 24 | 1024 | 4096 | 16 | 307M |

## A.11 IMPLEMENTATION DETAILS

### A.11.1 ARCHITECTURE

We use three variants of ViT transformer encoder backbone in our experiments. We show the configuration details and the number of parameters in Table 15. We also use hybrid model with ResNet50 as the tokenization backbone. We show the configuration in Table 16.

We use B-16 for the MLPMixer Tolstikhin et al. (2021).

We use two kinds of VQ Encoder in our experiment: Small VQ Encoder and VQ Encoder. The VQ encoder uses the same encoder architecture as the VQGAN encoder (Esser et al., 2021). For Small VQ Encoder, we decrease both the number of the resisual blocks and the number for filter channel by half, which is a lightweight model that decrease the model size by around 8 times. We show the configuration detail in Table 17.

## A.12 TRAINING

**Implementation details** We implement our model in Jax and optimize all models with Adam (Kingma & Ba, 2014). Unless specified otherwise, the input images are resized to 224x224, trained with a batch size of 4,096, with a weight decay of 0.1. We use a linear learning rate warm-up and cosine decay. On *ImageNet*, the models are trained for 300 epochs using three ViT model variants, i.e. Ti, S, and B, from small to large. The VQ-GAN model is also trained on ImageNet for 100K steps using a batch size of 256. The codebook size is $K = 1024$, and the codebook embeds the discrete token into $d_c = 256$ dimensions. For discrete ViT-B, we use average pooling on the final features. On *ImageNet-21K*, we train 90 epochs. We finetune on a higher resolution on ImageNet, with Adam and 1e-4 learning rate, batch size 512, for 20K iterations. The quantizer model is a VQ-GAN and is trained on ImageNet-21K only with codebook size $K = 8,192$. All models including ours use the same augmentation (RandAug and Mixup) as in the ViT baseline (Steiner et al., 2021). More details can be found in the Appendix.

*ImageNet Training* We train our model with the Adam (Kingma & Ba, 2014) optimizer. Our batch-size is 4096 and trained on 64 TPU cores. We start from a learning rate of 0.001 and train 300 epoch.

Table 16: Model configuration for ResNet+ViT hybrid models.

| Model | Resblocks | Patch-size | #Params |
|---|---|---|---|
| Hybrid-S | [3,4,6,3] | 1 | 46.1 |
| Hybrid-B | [3,4,6,3] | 1 | 111 |

Table 17: Model configuration for the VQ encoders.

| Model | Resblocks | Filter Channel | Embedding dimension | #Params |
|---|---|---|---|---|
| Small VQ Encoder | [1,1,1,1,1] | 64 | 256 | 3.0M |
| VQ Encoder | [2,2,2,2,2] | 128 | 256 | 23.7M |

We use linear warm up for 10k iterations and then a cosine annealing for the learning rate schedule. We use $224 \times 224$ resolution for the input image. We conduct experiment on three variants of the ViT model, Ti, S, and B, from small to large. In addition, we also experiment on the newly released MLPMixer, which also uses image patch as the token embeddings. We denote our approach as using both representations, and we also run a discrete only variant where we only uses the discrete embeddings without pixel embeddings. Due to the small input dimension of Ti, we only use the code representation without concatenate the pixel representation, and we project the 256 dimension to 192 via a linear projection layer. For the other variants, we concatenate both representations, and pad zeros if additional dimension required for the transformer. We use the same augmentation and model regularization as (Steiner et al., 2021), where we use RandAug with hyper-parameter (2,15) and mixup with $\alpha = 0.5$, and we apply a dropout rate of 0.1 and stochastic block dropout of 0.1. We download pretrained VQGAN model, which is trained on only ImageNet. We use VQ-GAN's encoder only with a codebook size of 1,024. The codebook integer is embedded into a 256 dimension vector. For the ViT-B discrete, we find training 800 epoch instead of 300 epoch can give another 0.1-1% gain over the test set, while training others for longer results in decreased performance.

*ImageNet-21K Training* Our models use learning rate of 0.001, weight decay of 0.03, and train 90 epoch. We use linear warm up for 10k iterations and then a cosine annealing for the learning rate schedule. We use $224\times224$ resolution for the input image. We use the same augmentation as (Steiner et al., 2021). For the quantizer model, we use VQGAN that is trained on unlabeled ImageNet-21K only. We use a codebook size of 8,192, and the codebook integer is embedded into a 256 dimension vector. To evaluate on the standard test set, we further finetune the pretrained ImageNet-21K model on ImageNet. We use Adam and optimize for 20k with 500 steps of warm up. We use a learning rate of 0.0001. We use a larger resolution $384 \times 384$ and $512 \times 512$. We only experiment the ViT-B variant.

For training VQ Encoder and Decoder, we train with batchsize 256 until it converges. We use perceptual loss with weight of 0.1, adversarial loss with weight 0.1. We use L1 gradient penalty of 10 during the optimization. The model is trained with resolution of $256 \times 256$. The model can scale to different image resolution without finetuning.

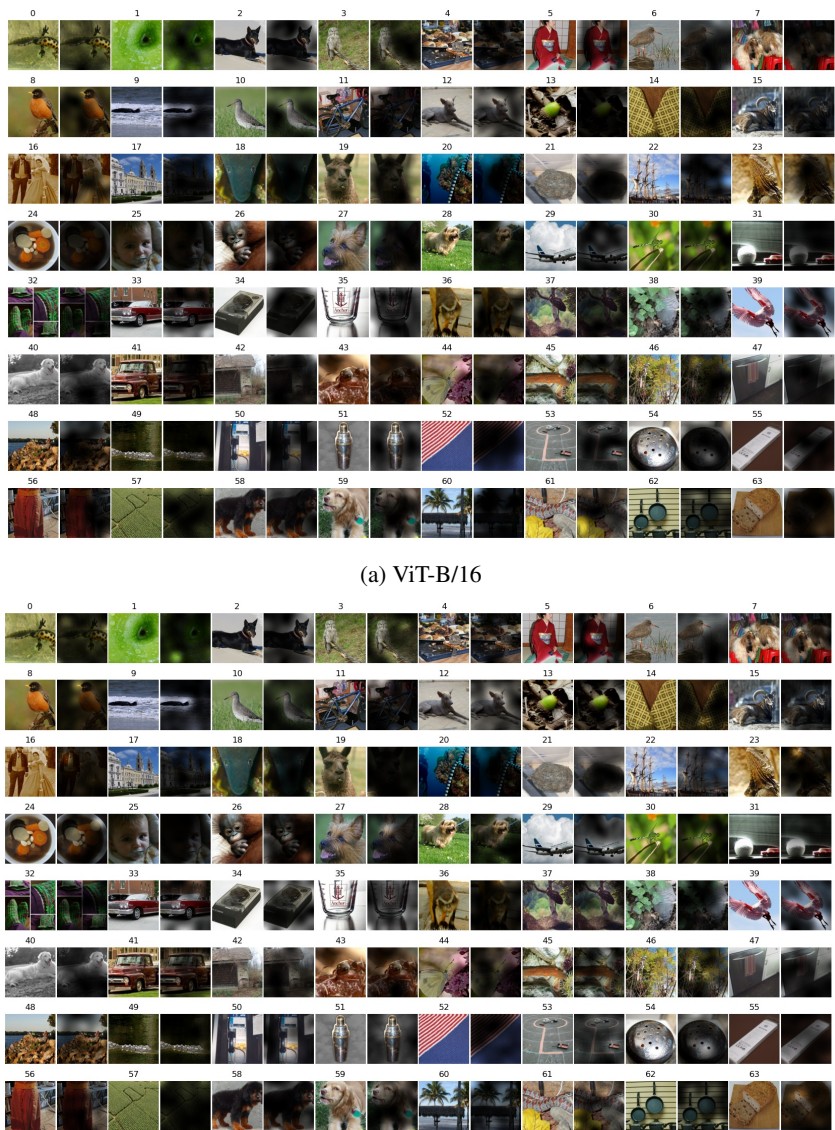

Figure 11: Attention comparison of the ViT and the proposed model on ImageNet 2012.

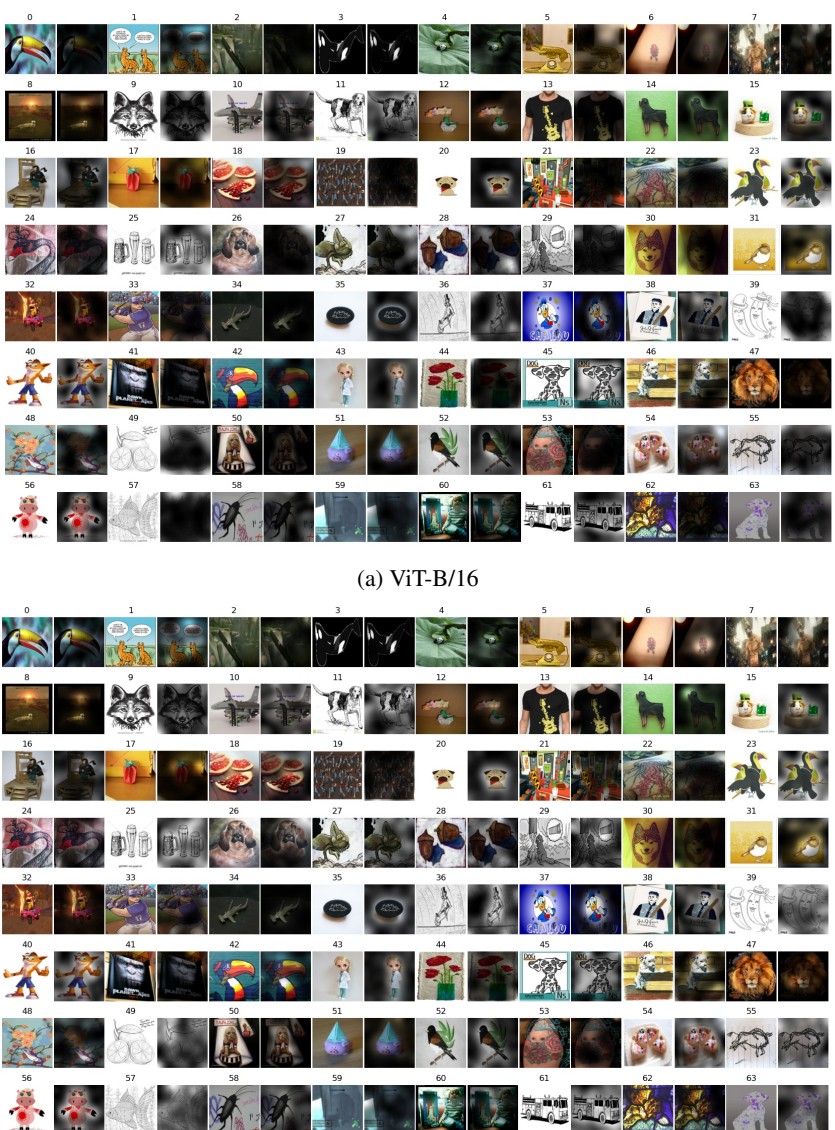

(a) ViT-B/16

(b) Ours ViT-B/16

Figure 12: Attention comparison of the ViT and the proposed model on ImageNet-R.

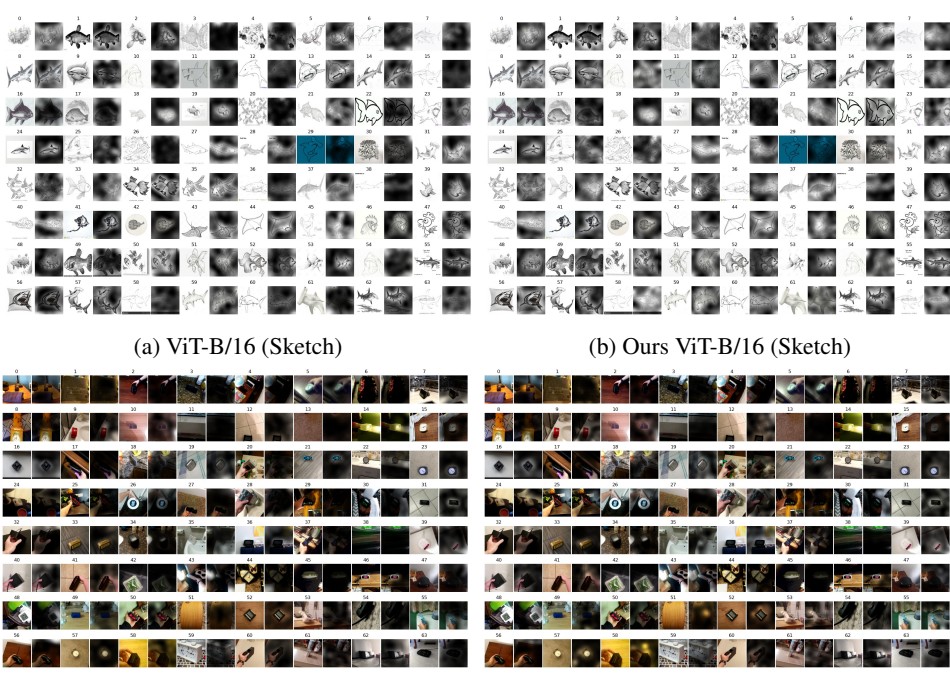

(a) ViT-B/16 (Sketch)                     (b) Ours ViT-B/16 (Sketch)

(c) ViT-B/16 (ObjectNet)               (d) Ours ViT-B/16 (ObjectNet)

Figure 13: Attention comparison of the VIT and the proposed model on ImageNet Sketch and ObjectNet.

