# OpenReview forum: "Discrete Representations Strengthen Vision Transformer Robustness"
_ICLR.cc/2022/Conference — ICLR 2022 Poster_

### Official Review · Reviewer_a2pv · 2021-10-22

**Correctness:** 3
**Technical Novelty And Significance:** 3
**Empirical Novelty And Significance:** 2
**Recommendation:** 5
**Confidence:** 4

**Details Of Ethics Concerns:**

No.

**Main Review:**

Pros.
1. The start point to use discrete information to handle robustness is novel.
2. The authors  provide some interesting opinions, such as "Using discrete tokens drives ViT towards better modeling of spatial interactions between tokens, given that individual tokens no longer carry enough information to depend on."
3. This paper is written distinctly, especially Introduction and Related work section.

Cons.
1. #Abstract: (1) The claim ViTs "are overly reliant on local features" rather than "make adequate use of global context" leads ViTs to fail to generalize OOD, real-world data. #Introductioin: (2) ViTs's robustness comes from capturing globally-contextualized inductive bias than CNNs. (3) Spatial structure contributes less than 3% of ViT's performances. The (1)(2)(3) seems to be contradictory.
2. The number of baselines is very limited. Missing many important methods, such as CutMix, Mosaic, Puzzle Mix, Cutout, Manifold.
3. The reviewer is confused about whether the global information is the key weapon of your model or the discrete information.
4. Missing enough discussion of the method itself. (1) After reading the paper, the reviewer still gets less information on the "codebook". (2)  Is VQ-GAN the chosen one, how about using other technologies, and so forth.
5. Missing some related works such as Pyramid Vision Transformer: A Versatile Backbone for Dense Prediction without Convolutions; Twins: Revisiting the design of spatial attention in vision transformers.

Others issues.
1. In Figure 3(b), the author noticed that the "Ours" model captures more structural and shape patterns. And in 4.2 Qualitative results, authors hypothesize the robustness improvement stems from better capture of global contexts. So is that mean:  the ViT (or related models) learns higher frequencies is bad?
2. Why using discrete representation can improve robustness? However, such an operation results in information loss. If it works, how about directly concat a global feature extracted by CNN?

**Summary Of The Paper:**

This paper introduces Dr. ViT, which is conducted on 7 ImageNet robustness benchmarks. The motivation of this work is to demonstrate the robustness can be enhanced when adding discrete representation. Sufficient experiments demonstrate the effectiveness of this paper.


**Summary Of The Review:**

Suggestion.
1. The authors organize splendid discussions in the introduction and related work section, and some discussions look interesting. But the latter paragraphs could not work well in concert with the previous writing.  It is suggested that supplement enough in-depth analysis to demonstrate the mentioned illustrations. After reading this paper, the reviewer could just observe "ViT + some global information would be better" rather than your method is necessary.

---

> ### Author Response · Authors · 2021-11-23
> **Thank you for the review. We added all suggested experiments (Part 1)**
>
> We thank the reviewer for their valuable feedback. We appreciate the reviewer’s assessment that *“discrete information to handle robustness is novel”* and *“the authors provide some interesting opinions”*. We add all the suggested experiments and address the reviewer’s questions.
>
>
> **Abstract(1) and Introduction (2,3) seem to be contradictory.**
>
> We apologize for the confusion. The issue is between (2) and (3), both of which are from the related work. We have updated the introduction to clarify this (see the text in blue), and here is the sketch:
>
> - (2) is from the related work, where they suggested ViT improves robustness than CNN because attention architecture can capture global context better.
> - However, (3) is from a more recent work [1] which shows that ViT trained on ImageNet may not exploit the global context as expected.
> - Our statement (1) follows (3), where we substantiate the claim (1) by the modified Section 4.3 showing discrete representation can better capture global context.
>
> [1] Xinlei Chen, Saining Xie, and Kaiming He.  An empirical study of training self-supervised vision transformers. arXiv preprint arXiv:2104.02057, 2021b
>
>
> **Adding more baselines.**
>
> Following the suggestion, we ran 6 additional baselines with the recent data augmentation methods in Appendix A.3, including CutMix and CutOut. In Table 7, our method also outperforms these baselines on the robustness benchmark.
> Note that our method is conceptually different and complementary to these data augmentation methods. We leave incorporating more advanced augmentations in our future work.
>
> **Whether the robustness comes from the global information captured by the tokenization model or the discrete information.**
>
> Following the suggestion, we add a new baseline that directly concatenates the continuous global feature extracted by CNN to the ViT’s input token. The only difference between this baseline and ours is that ours concatenates discrete embeddings rather than CNN global features. We extensively search the concatenation dimension in Appendix A.4, and summarize the best model below. Concatenating global CNN features does not improve ViT’s robustness. More results are shown in Appendix A.4.
>
> |  Model   | ImageNet | Real | Rendition |Stylized| Sketch|ObjectNet|V2|A|C &darr;|
> |--|--|-|-|-|-|-|--|-|--|
> |ViT|78.73 | 84.85 | 38.15 | 10.39 | 28.60 | 28.71 | 67.34 | 16.92 | 53.51 |
> |ViT+Global CNN Continuous Feature| 78.51 | 83.44 | 34.66 | 10.47 | 25.14 | 28.35 | 65.86 | 15.97 | 57.64|
> ViT + Ours | **79.48** | **84.86** | **44.77** | **19.38** | **34.59**  | **30.55**  | **68.05** | **17.20** | **46.22** |
>
> It seems to be a misunderstanding that our method is better because it brings useful global information to the ViT model. We note that the self-attention in ViT is already very capable of modeling global dependency. Further adding global features extracted from CNN would not improve this capability. On the contrary, discrete representations carry semantics learned in the VQ-VAE/VQ-GAN to facilitate ViT to better capture object shape and global context. We have substantiated this claim by the extensive experiments in Section 4.3.
>
> **Why does using discrete representations improve robustness?**
>
> As pointed out by Reviewer 2 (mNoS), discrete representations from VQ-GAN can improve ViT robustness because “they preserve shape and structure object information”. As we have shown in Figure 2, discrete representation discards nuisances and keeps the semantic information. Consider the flamingo images in Figure 2. The reconstructed images keep the flamingo shape but interrupt texture details such as the word “flamingo”, watermark, and the background that are unimportant for flamingo recognition. Losing nuisance details forces the model to recognize images not just by textures but also by shapes and contexts. Our experiments in Section 4.3 substantiate this claim.
>
> **Add an experiment that concatenates a global feature extracted by CNN.**
>
> We conducted the experiment suggested by the reviewer and showed more results in Appendix A.4. Direct concatenating a global, continuous feature extracted by CNN does not improve robustness, which verifies the necessity of the discrete representation.
>
> **Discussion on Codebook.**
>
> We have conducted four studies to analyze the codebook, and include the discussions on
>
> -  the effect of the codebook size (Table 6 and the second paragraph in Section 4.4)
> -  the impact of codebook encoder capacity in Appendix A.6
> -  the effect of codebook with more data in Appendix A.7
> -  comparing codebooks learned by VQ-GAN and VQ-VAE in Appendix A.8.
>
> **Related work**
>
> Thank you for the suggestion. We have cited the missing references in the related works of our revision.

---

> > ### Author Response · Authors · 2021-11-23
> > **Thank you for the review. We added all suggested experiments (Part 2)**
> >
> > **Justification for using VQ-GAN and other technologies for Discrete Representation.**
> >
> > We use VQ-GAN because it is the state-of-the-art method vector quantization model for images. Following the suggestion, we also experiment with the VQ-VAE model which is an older model for image quantization.
> > We show the key VQ-VAE results In the table below and more results in Table 11, where reasonable improvements are obtained over the ViT baseline. Specifically, VQ-VAE improves the robustness of ImageNet-Rendition by 8%, Stylized-ImageNet by 13%, etc. This experiment validates VQ-GAN is not the chosen one, which suggests our model is compatible with other image quantization models.
> >
> > |  Discrete Tokenization Model   | ImageNet | Real | Rendition |Stylized| Sketch|ObjectNet|V2|A|C &darr;|
> > |-|--|-|-|-|-|-|-|--|---|
> > |None|78.73 | 84.85 | 38.15 | 10.39 | 28.60 | 28.71 | 67.34 | 16.92 | 53.51 |
> > |VQ-VAE |78.36 | 84.35 | **46.22** | **23.36** | **35.17** | **29.34** | 66.24 | 13.61 | **52.20** |
> > VQ-GAN | **79.48** | **84.86** | **44.77** | **19.38** | **34.59**  | **30.55**  | **68.05** | **17.20** | **46.22** |
> >
> >
> > **Is it bad that the ViT (or related models) learns higher frequencies?**
> >
> > As shown in Table 1, capturing high frequencies helps improve in-distribution performance on ImageNet, but it might undermine OOD generalization. In addition, some higher frequencies can be useful for humans, such as the edge information in ImageNet-Sketch, which our discrete model also captures (e.g., the flamingo sketch in Figure 2).
> >
> >
> > **Supplement enough in-depth analysis to demonstrate the mentioned illustrations**
> >
> > We made two claims in the paper 1) discrete representations strengthen ViT robustness, and 2) discrete representations facilitate ViT to better capture object shape and global contexts. We have demonstrated the former by extensive experiments training ViTs trained both on ImageNet and the large-scale ImageNet-21K and evaluating on seven robustness benchmarks.
> >
> > We modified a new Section 4.3  to demonstrate the second claim about shape and global context. Here is the sketch of Section 4.3:
> >
> > We made two claims in the paper 1) discrete representations strengthen ViT robustness, and 2) discrete representations facilitate ViT to better capture object shape and global contexts. We have demonstrated the former by extensive experiments training ViTs both on ImageNet and the large-scale ImageNet-21K and evaluating on seven robustness benchmarks.
> >
> > We modified a new Section 4.3  to demonstrate the second claim about shape and global context. Here is the sketch of Section 4.3 showing our Discrete ViT (ViT with discrete representations):
> >
> > * Quantitative results
> >     - Discrete ViT improves ViT’s accuracy in recognizing object shape on Stylized-ImageNet by 10+%.
> >     - Discrete ViTs are 20% more shape-biased than ViT and and 40% more than ResNet.
> >     - Position embedding is essential for Discrete ViTs, which is not the case for the standard ViT.
> > * Qualitative results
> >     - Filters of Discrete ViT capture more structural patterns.
> >     - Attentions of Discrete ViT are more global and relevant to the object.
> >     - Decoded images from discrete representations reasonably preserve shape and structure information.

---

> > > ### Author Response · Authors · 2021-12-06
> > > **Follow-up**
> > >
> > > Thank you again for your thoughtful review and suggestions. We have included all the experiments that you suggested. Would you kindly increase the numerical scores if our rebuttal has addressed your questions? Also, please let us know if you have any questions.

---

### Official Review · Reviewer_mNoS · 2021-11-02

**Correctness:** 3
**Technical Novelty And Significance:** 4
**Empirical Novelty And Significance:** 4
**Recommendation:** 8
**Confidence:** 4

**Main Review:**

Strengths:
1. The proposed method share a similar idea with BEiT using discrete VAE to "tokenize" image while apply the idea to improve ViT in different ways. To the best of my knowledge, concatenating discrete representation with the continuous representations for ViT has not been studied in previous work.
2. The proposed method is technically sound and the writing is properly formatted, well organized and easy-to-follow.
3. The proposed method is simple yet effective as shown in the experiments. It is a general approach to improve the robustness of ViT. The experiments show that the approach consistently outperforms baseline methods on the benchmarks and achieve SoTA on ImageNet-Rendition, Stylized-ImageNet and ImageNet-Sketch.
4. The authors present the observation that discrete representations derived from VQ-VAE preserve shape and structure object information which can be integrated into ViT to improve the robustness. They also conduct insightful experiments to demonstrate this observation qualitatively. The quantitative evaluation on the ImageNet benchmarks seems to support this observation.
5. The authors present extensive ablation studies as listed below. The results are convincing.
 - As the proposed method capture shape information, positional embeddings play important role for their proposed models comparing to plain ViT
 - the improvement does not trivially come from using larger models
 - analysis on codebook size is reasonable and outcome is consistent to ones would expect.


Weaknesses:
1. In Figure 3(b). the visualization shows that discrete representations promote ViT to learn structured filters. Why the proposed modification is able to achieve this behavior is unclear. According to the rest of the paper, the discrete representation itself carries structure information to improve the robustness. Ideally the filters should pick up the complementary information that are more related to style and texture.
2. There are high attention scores on the corners for ViT In Figure 4 (ii). Why is this the case?
3. The description of the optimization on \phi in equation (7) is not precise. A term E_q(z|x)[logp(y|x,z)] can be factored out from the KL term and the gradient of this term w.r.t. \phi seems not considered in learning VQ_VAE. As a result, the optimization is not exactly maximizing the ELBO described in the equation. However, this term seems impossible to address using pre-training and fine-tuning approach given that p(y|x, z) is not properly modeled during pre-training.
4. The comparison to the SoTA methods presented In Table 3 might be unfair. The proposed method is based on ViT-B (86M parameters) which is bigger than ResNet-50 (25M parameters) on which the other competing methods are based. The competing methods might achieve similar performance using ViT-B. Some of the improvement might come from a bigger model and it would be great if the authors can clarify this and add the discussion in the paragraph.
5. Minor issues:
 - Notation L seems used to denote two different things in Section 3.1. It denotes the length of the sequence in the third sentence (L=HxW/P^2) while it also denotes number of transformer layers in equation (2) and (3).
- In the paragraph above equation (6), z_d takes value from {1, 2, .., K}. Maybe it is better to write z_d \in {1, 2, ..., K}^L.
 - In the paragraph above section 4.3, the authors refer ImageNet-Rendition as ImageNet-R. It would be great to stay consistent using ImageNet-Rendition as ImageNet-R only appears here.


**Summary Of The Paper:**

The author present an observation in this paper that discrete image token representations derived from a vector quantized image encoder is able to preserve shape and structure object information. Inspired by this observation, the authors propose a modification to ViT architectures that appending these token representations to the input and show that the resultant models generalize better for out-of-distribution data on ImageNet classification. The modification to the input is simple and can be integrated into variants of ViT. Experiments show that the proposed method consistently outperforms baseline models on ImageNet, ImageNet-Real and seven out-of-distribution datasets derived from ImageNet. The margin of improvement is especially larger for the tasks where textures and image style change significantly while object structure is more discriminative. The main contributions of this paper are two-fold: 1) a novel approach to enrich existing ViT architectures with shape and structure information derived from discrete representations and 2) improve the robustness over the baseline ViT models and achieve SoTA results on ImageNet-Rendition, Stylized-ImageNet and ImageNet-Sketch.


**Summary Of The Review:**

The submission is pretty solid and the claims made in the paper are convincing. To the best of my knowledge, the proposed method is novel and the contributions are significant. The authors conduct detailed ablation study which are pretty interesting to learn. There are some observations unclear to me, but they are not critical issues given the solid experiments presented in the paper. Overall I believe the quality of this submission is great.

---

> ### Author Response · Authors · 2021-11-23
> **Thank you for the detailed review.**
>
> We thank the reviewer for their detailed and insightful feedback. We are glad that the reviewer found our idea novel, technically sound, and our results convincing. We address the reviewer's question below.
>
> **Reason for learning structured filters for pixel token in ViT.**
>
> Thank you for the question. We agree with the reviewer that the filters can pick up complementary information. In Figure 3(b), we visualized our default model after tuning the pixel-dimension in Table 13. Our best-performing model uses pixel-dimension=32, which appears to introduce an inductive bias that encourages the filters to learn more structured information. We provide more visualizations in Figure 9 by increasing the pixel-dimension. More discussions are in Appendix A.1.3.
>
>
>
> **High attention scores on the corners for ViT.**
>
> This seems to be an inherent property of ViT, which was also observed in other works, for example in Figure 4 of the paper [Intriguing Properties of Vision Transformers](https://arxiv.org/abs/2105.10497), such as the “Layer 01” row and the  “Layer 12” row.
>
> **Optimization on \phi in Equation (7).**
>
> We really appreciate the useful comments. In the revision, we have made our assumption very clear that we assume a uniform prior for both P(y|x,z) and P(z). With this assumption, we believe our first stage now maximizes the ELBO. We also briefly discuss the work our assumption is inspired from, and the stability advantage to not learning ViT parameters in the first stage.
>
>
> **Discussion on the comparison to the SoTA methods.**
>
> Thank you for the suggestion. We have clarified the comparison to the STOA should consider the backbone difference. In addition, we add more baselines (Cutout and CutMix) trained using the same ViT backbone.
>
> **Minor issues.**
>
> Thank you for pointing them out. We have fixed them in the revision.

---

### Official Review · Reviewer_rENv · 2021-11-02

**Correctness:** 2
**Technical Novelty And Significance:** 2
**Empirical Novelty And Significance:** 2
**Recommendation:** 3
**Confidence:** 4

**Main Review:**

Authors state that ViTs are overly reliant on local features and fail to make adequate use of global context. To me, this is counter-intuitive and is not supported by concurrent recent findings.

Inspired by https://openreview.net/forum?id=Bygh9j09KX that shows CNNs are biased towards texture; [1] conducts an extensive study, and demonstrates that ViTs are less biased towards texture, compared with their CNN counterparts. The authors show that ViTs have shape-bias, comparable to humans. See Fig5 of the paper.

[1] “Intriguing Properties of ViTs” Neurips’21 https://arxiv.org/pdf/2105.10497.pdf

To make a claim on texture vs Shape bias, the authors must conduct a principled study as in “ImageNet-trained CNNs are biased towards texture; increasing shape bias improves accuracy and robustness”  https://openreview.net/forum?id=Bygh9j09KX

In recent works, ViTs demonstrate better out-of-domain generalization, as demonstrated in extensive experiments conducted in Fig14 of https://arxiv.org/pdf/2105.10497.pdf  and Fig5 of  https://arxiv.org/abs/2106.09785


**Summary Of The Paper:**

Authors propose discrete tokens (instead of the standard continuous pixel-values projected and fed as tokens to ViTs), to enhance the shape learning capability of ViTs, and thus make them robust against out-of-distribution data. The underlying assumption is that discrete tokens preserve the global structure and lose local details, and are therefore better for robustness.


**Summary Of The Review:**

In the paper, the experiments are conducted primarily on ImageNet robustness datasets. It will be interesting to see, out-of-domain generalization of the features learned by the proposed approach e.g., ImageNet pre-trained features on fine-grained datasets (Flowers, Pets, Fungi, Sketches, Birds, etc).

I am not entirely convinced by the shape-vs-texture bias claims made in the paper, due to lack of principled study as in  https://openreview.net/forum?id=Bygh9j09KX

Also, it will be interesting to see if the proposed discrete tokens enhance ViTs robustness against common corruptions (e.g., artifacts introduced by rain, haze etc), and adversarial perturbations. The considered robustness analysis is too restrictive in my opinion.

---

> ### Author Response · Authors · 2021-11-10
> **Thank you for the review. Clarification to the misunderstanding and confusion**
>
> **Q1: The authors show that ViTs have shape-bias, comparable to humans. See Fig5 of the paper [1] https://openreview.net/forum?id=Bygh9j09KX.**
>
> We respectfully point out that it is a misunderstanding. In [1], the authors trained the ViT on the Stylized-ImageNet and evaluated it on the held out Stylized test set (called SIN-SIN). We note this is different from our out-of-distribution generalization setup (which is IN-SIN), where the model has to be tested on a distribution that it has never been trained on before. By contrast, we follow the standard setting to train the model on ImageNet or ImageNet-21k and evaluate the model on seven out-of-distribution benchmarks derived from ImageNet.
>
> **Q2: Authors state that ViTs are overly reliant on local features and fail to make adequate use of global context. To me, this is counter-intuitive and is not supported by concurrent recent findings.**
>
> Our statement that ViT (trained on ImageNet) is consistent with recent findings. In our Table 1, we showed ViT performs better than CNN in robustness generalization which is consistent with the existing finding in [1].
>
> In our Section 4.3 (Position embedding: local vs. global context), we showed removing position embedding (which removes global structure) only led to a marginal performance drop, suggesting that ViT does not make adequate use of global context. This is consistent with recent findings, for example:
> > “However, we also find that the accuracy only decreases by a bit even if removing the only positional inductive bias (position embedding), suggesting that in our method ViT relies less on positional information” [2]
>
> also supported by [1]
>
> > “the effect of positional encoding towards injecting structural information of images to ViT models is limited”[1]
>
> In the same Section 4.3, our model using discrete tokens performances 30-80% worse after removing the position information, substantiating that the discrete tokens makes better use of the global information.
>
> [1] [Intriguing Properties of ViTs](https://arxiv.org/pdf/2105.10497.pdf) In Neurips’21
>
> [2] [An empirical study of training self-supervised vision transformers](https://arxiv.org/abs/2104.02057) In ICCV’21
>
>
> **Q3: To make a claim on texture vs Shape bias, the authors must conduct a principled study as in [1] https://openreview.net/forum?id=Bygh9j09KX**
>
> Thanks for the comment but we have performed a key study as in [1] (https://openreview.net/forum?id=Bygh9j09KX). An important study on shape-bias is the Stylized-ImageNet in Table 1 of [1]. Since our topic is to improve the robustness, the setting relevant to ours is called IN-SIN (trained on ImageNet and tested on Stylized-ImageNet). We performed the same experiment in our paper (c.f. “Stylized” Column in Table 1, Table 2, and Table 3(c)). In particular in Table 3(c), our baselines BagNet and Resnet50 were taken from the IN->SIN column in Table1 from the paper the reviewer mentioned. We achieve 10%-20% improvement over [1] on generalization on Stylized-ImageNet, demonstrating our method to achieve significantly better robustness towards shape-bias.
>
>
> **Q4: In recent works, ViTs demonstrate better out-of-domain generalization, as demonstrated in extensive experiments conducted in Fig14 of https://arxiv.org/pdf/2105.10497.pdf and Fig5 of https://arxiv.org/abs/2106.09785**
>
> We believe the reviewer confused transfer learning with out-of-distribution generalization. The Fig14 of https://arxiv.org/pdf/2105.10497.pdf and Fig 5 of https://arxiv.org/abs/2106.09785 are measuring the feature transfer on standard “transfer” learning datasets, which requires the model to be trained on the target distribution and tasks. On the other hand, domain generalization only tests on the new distribution without training on that.
> We agree transfer learning is an interesting setting but is not in the scope of our paper.
>
> **Q5: it will be interesting to see if the proposed discrete tokens enhance ViTs robustness against common corruptions (e.g., artifacts introduced by rain, haze etc), and adversarial perturbations. The considered robustness analysis is too restrictive in my opinion.**
>
> We respectfully point out that we have already validated the robustness against the 15 types of common corruptions in ImageNet-C [4]. See Table 4 in our paper, which includes corruptions such as Fog (Haze), Snow, Frost, etc. We improved upon the ViT model trained on ImageNet-21K by 7% on Fog, 9% on Frost, 5% on Snow. We think the improvement is significant given that we improve constantly across all 15 common corruptions by an average of 10% than the state-of-the-art ViT model.
>
> Our results also include ImageNet-A [5] which is about natural adversarial images. We will make this clear in our revision
>
> [4] [Benchmarking neural network robustness to common corruptions and perturbations](https://arxiv.org/abs/1903.12261) ICLR 2019
>
> [5] [Natural adversarial examples](https://arxiv.org/abs/1907.07174). CVPR, 2021b.

---

> > ### Author Response · Authors · 2021-11-23
> > **We sincerely hope the review can engage in the discussion.**
> >
> > **We sincerely hope the review can engage in the discussion and respond to our comments.**
> >
> > **Principled study on texture vs. Shape bias.**
> >
> > We performed the same shape-bias study as in [1] and [2] as suggested by the reviewer.  The detailed results are in the “quantitative results” paragraphs of Section 4.3. Below we summarize the key findings:
> >
> > **Study on shape-bias**: We quantify the shape-bias as the fraction of shape decisions, and present the results in Figure 5(a) of our paper which follows the convention of Figure 4 in [1] and Figure 6 in [2]. We consider the only relevant out-of-distribution generalization setting “IN-SIN”, i.e. training models on ImageNet and testing on Stylized-ImageNet. The table below is from Figure 5(a) of our paper:
> >
> >
> > | Model | Shape bias (fraction of shape decisions) |
> > |---|---|
> > | ResNet-50 | 0.22 |
> > | ViT | 0.42 |
> > | ViT+ CNN Global Features | 0.3 |
> > | Discrete ViT (Ours) | **0.62** |
> > —
> > | Humans | 0.96 |
> >
> > As shown, ViT (0.42) is more shape-biased than CNN ResNet-50 (0.20), which is consistent with the prior work [2]. Adding discrete representation (0.62) greatly shrinks the gap between the ViT (0.42) and human baseline (0.96), demonstrating our model has more shape-bias than the ViT. In fact, our model has the highest shape-bias (0.62) among all the vision transformer models in Figure 6 of [2] that are not trained on the Stylized ImageNet (SIN).
> >
> > **Study on shape recognition**: In addition, following the texture vs. shape study as in Table 1 of [1], we have shown that our model outperforms the state-of-the-art shape recognition accuracy on the Stylized-ImageNet. The table below is from the Table 3 (c) of our paper:
> >
> > | Model | Top-5 on Stylized ImageNet |
> > |---|---|
> > | BagNet-33 | 4.2 |
> > | ResNet-50 | 16.4 |
> > | ViT-B | 22.2 |
> > | ViT-B + CutOut | 24.7 |
> > | ViT-B + CutMix | 22.7 |
> > | ViT-B *21K | 31.3 |
> > | Our ViT-B (discrete only) | **40.3** |
> >
> > The above two studies cover all relevant key studies on shape vs. texture in [1]. The results show discrete representations facilitate ViT better capturing object shape and global contexts.
> >
> > [2] [Intriguing Properties of ViTs](https://arxiv.org/abs/2105.10497) In NeurIPS 2021
> >
> > [1] [ImageNet-trained CNNs are biased towards texture; increasing shape bias improves accuracy and robustness](https://openreview.net/forum?id=Bygh9j09KX). In ICLR 2019.

---

> > > ### Author Response · Authors · 2021-12-06
> > > **Follow-up**
> > >
> > > Thank you again for your thoughtful review and suggestions. Following your suggestions, we have conducted experiments [1] in Figure 5(a) to support the "shape-vs-texture bias claims".  Reviewer YfcP thinks the new result in Figure 5(a) support the claim. Would you kindly increase the numerical scores if our rebuttal has addressed your questions?
> > >
> > > [1] Geirhos et al. ImageNet-trained CNNs are biased towards texture; increasing shape bias improves accuracy and robustness. ICLR 2019.

---

### Official Review · Reviewer_YfcP · 2021-12-03

**Correctness:** 2
**Technical Novelty And Significance:** 3
**Empirical Novelty And Significance:** Not applicable
**Recommendation:** 8
**Confidence:** 3

**Main Review:**

Pro:
- How to improve the robustness of ViTs is rarely studied in the literature. The authors proposed a novel approach to improve the ViT's robustness, and made a comprehensive study on it.
- The paper is well written.
- Results showed the effectiveness of the proposed approach on improving the robustness of ViTs.

Con:
- The authors claimed that ViTs' fail to make adequate use of global context. I am not entirely convinced by this claims. From results of the paper, I believe that the proposed method can improve the robustness of ViTs, but it can not support the claim. The experiment of discarding position embedding can only prove that positional information contribute small information for the classification. It risk to claim that this observation shows that ViTs do not fully use the global context information.

**Summary Of The Paper:**

The authors have an observation that ViTs trained on ImageNet heavily depend on local features, but fail to use the global features (shape or structure). To address this issue and improve the robustness of ViTs, the authors proposed to replace the linear embedding layer by the a vector-quantized encoder. The authors claims that it can push the ViTs to learn the global information by this replacement. Experiments are conducted on ImageNet and other ImageNet variant datasets. Results show that the proposed method can improve ViTs' robustness on various benchmarks.

**Summary Of The Review:**

The authors proposed an novel approach to improve the ViT's robustness. Comprehensive experiments are conducted on various robustness benchmark. Results showed the proposed method can improve the robustness of ViTs. My main concern is that the major claim of the paper, that ViT's fail to make adequate use of global information, is not supported by the experiments. I'd like to accept the paper if the authors can make the claim more supportive.

---

> ### Author Response · Authors · 2021-12-04
> **Thank you for your thoughtful review. We will revise the paper.**
>
> Thank you for your thoughtful review. We are glad that you found our paper to be novel, effective, well-written, and have a comprehensive study.  We think you make good points. We will revise the paper in the following ways:
>
> * We will revise the claim. We agree that the term “global information” is not well defined. We will revise that term to be “shape information”, which is well defined in [1]. "Shape information" is directly supported by our experiments. Our Figure 5(a) follows the established cue conflict experiment in paper [1] to evaluate the models' shape bias, and we show that discrete representations learn shape bias 20% more than continuous representations, and thereby improving robustness.
>
> * Position embedding is important empirically, but we agree with the reviewer that it is not clear how that directly relates to the global context. Since our paper is about discrete representation, position embedding is not the major claim of the paper. We will remove the part that relates to position embedding. Note that this removal does not impact our paper as our major claim is on discrete representation and robustness, not positional embedding. We receive this review after the revision period, due to the ICLR policy, we cannot update the paper at this point. But this is a small change for us, which we promise to update in the revision.
>
> [1] Geirhos et al. ImageNet-trained CNNs are biased towards texture; increasing shape bias improves accuracy and robustness. ICLR 2019.

---

> > ### Comment · Reviewer_YfcP · 2021-12-04
> > **Follow-up**
> >
> > Thank you for the response. The authors has addressed my main concern. I lean to accept the paper.

---

> > > ### Author Response · Authors · 2021-12-04
> > > **Thank you for your quick response.**
> > >
> > > Thank you for your quick response! We are glad to hear that the main concern has been addressed and you now lean to accept the paper. Could you please also modify the numerical score to match the written review? Also, please let us know if you have other questions.

---

### Decision · Program_Chairs · 2022-01-20

**Decision:**

Accept (Poster)

**Comment:**

Summary: Authors present an approach to improve the robustness of vision transformers by mapping standard tokens into discrete tokens that are invariant to small perturbations. Method is applied to a variety of backbone architectures and evaluated on a range of out of distribution forms of ImageNet test set. Significant performance gains are measured across many of these tasks.

Pros:
- Novel, simple, effective approach
- General approach applicable across model variants, complimentary to other methods to improve robustness.
- Comprehensive study, evaluated on many ImageNet robustness benchmarks
- Well written overall

Cons:
- Biggest issue: 3 reviewers point out concerns about validity of claims that ViT architecture is more reliant on local patterns and less on global context. This seems mostly a semantic issue around conjectures about why the method works – it does not invalidate the value of the new approach or its solid results.  Authors have responded to reviewer concerns by changing wording in paper to relax the claims, specifying “shape information” rather than “global information”.  They have also added experiments to measure shape bias, as defined in prior art, to backup these claims.
- Paper missing baselines of data augmentation strategies. Authors have responded by including such comparative experiments.
- Paper is missing ablation studies on changing the type of codebook. Authors have responded by including multiple variations of codebooks, and varying the codebook size.

This paper was a close call based on the reviews. However, in AC opinion, the critiques have been adequately addressed by the authors. This is confirmed by adding an extra expert reviewer to the pool, who agreed with some earlier critiques, and was satisfied with the changes and additional experiments presented by the authors. AC recommendation is to accept.